# Industrial source identification of polyhalogenated carbazoles and preliminary assessment of their global emissions

Yuxiang Sun[1,2,3], Lili Yang[2,3], Minghui Zheng [1,2,3], Roland Weber[4], Jerzy Falandysz[5], Gerhard Lammel [6,7], Chenyan Zhao[1,2,3], Changzhi Chen[1,2,3], Qiuting Yang[2,3] & Guorui Liu [1,2,3]

Polyhalogenated carbazoles (PHCZs) are emerging global pollutants found in environmental matrices, e.g., 3000 tonnes of PHCZs have been detected in the sediments of the Great Lakes. Recognition of PHCZ emissions from ongoing industrial activities worldwide is still lacking. Here, we identify and quantify PHCZ emissions from 13 large-scale industries, 12 of which previously have no data. Congener profiles of PHCZs from investigated industrial sources are clarified, which enables apportioning of PHCZ sources. Annual PHCZ emissions from major industries are estimated on the basis of derived emission factors and then mapped globally. Coke production is a prime PHCZ emitter of 9229 g/yr, followed by iron ore sintering with a PHCZ emission of 3237 g/yr. China, Australia, Japan, India, USA, and Russia are found to be significant emitters through these industrial activities. PHCZ pollution is potentially a global human health and environmental issue.

PHCZs, as emerging dioxin-like compounds (DLCs)[1] (structures provided in Supplementary Fig. 1), have been detected in soil[2,3], sediment[4,5], biota[5-7], air[8], and water[9-11] in more than 20 regions of Asia, North America, and Europe. Their concentrations are comparable to or even exceed those of some traditional DLCs. The total concentration of 11 PHCZs in Lake Tai is 1.54 ng/g[4], which is close to that of polychlorinated biphenyls (PCBs) and polybrominated diphenyl ethers (PBDEs). PHCZs with concentrations of up to 13 ng/g were found in soils from an e-waste dismantling areas[12], where polychlorinated dibenzo-p-dioxins and polychlorinated dibenzo-p-furans (PCDD/Fs), as well as PBDEs, were found at 3.23 ng/g and 1.91 ng/g. An investigation of the pollution status of the Great Lakes revealed the existence of 26 PHCZs with a median concentration of 23.7 ng/g in surface grabs, leading to an estimated >3000 tonnes in sediment, which is orders of magnitude greater than those of PCBs and PBDEs[13-15].

Fungal activity can be a natural PHCZ source. Mumbo et al.[16] confirmed that in the presence of $H_2O_2$ and chloride and bromide ions, fungi can bio-transform carbazole (CZ) to PHCZs. The load of 1,3,6,8-BCZ and other 12 unknown PHCZs constituting 64% of more than 3000 tonnes of PHCZ residues in sediments from the Great Lakes is concluded to be generated from natural activity, implying on the significant natural formation of PHCZs[14]. It is widely believed that anthropogenic activities also play a role in PHCZ pollution of the environment[4,5]. Recently, PHCZs have been found in the disinfection of drinking water[10]. The production of halogenated indigo dyes[17] and specific photoelectric materials[18] are also considered potential sources of PHCZs. 2,7- and 3,6- halogenated carbazoles are possible intermediates of two polymers present in electronic devices, which have not been detected directly in these industrial samples. The synthesis of halogenated indigo dyes is also considered a source of PHCZ

[1]School of Environment, Hangzhou Institute for Advanced Study, UCAS, Hangzhou 310024, China. [2]State Key Laboratory of Environmental Chemistry and Ecotoxicology, Research Center for Eco-Environmental Sciences, Chinese Academy of Sciences, Beijing 100085, China. [3]College of Resource and Environment, University of Chinese Academy of Sciences, Beijing 100049, China. [4]POPs Environmental Consulting, Lindenfirststr. 23, 73527 Schwäbisch Gmünd, Germany. [5]Medical University of Lodz, Faculty of Pharmacy, Department of Toxicology, Muszyńskiego 1, 90-151 Łódź, Poland. [6]Max Planck Institute for Chemistry, Mainz 55128, Germany. [7]RECETOX, Faculty of Science, Masaryk University, 60177 Brno, Czech Republic. ✉e-mail: grliu@rcees.ac.cn

emission[19,20], although one modern indigo dye, 5,5',7,7'-tetra-bromoindigo was proved not to contain PHCZs[19]. Only 1,3,6,8-CCZ, 1,3,6,8-BCZ, and 1,8-B-3,6-CCZ were the main impurities in historical synthetic dye[19]. The postulation that industrial activities unintentionally releases PHCZs is generally based on the decrease in PHCZ concentration over time when relevant activities cease, distance-dependent distribution patterns around factories, and the correlation of environmental levels with the activities of surrounding industries in some specific areas[21–23]. Thus, current research on source identification is woefully inadequate and does not support any reasonable explanation of the abundant residues, wide distribution, and complicated congener profiles of PHCZs in the environment[14]. Artificial sources of PHCZs remain unresolved but urgently need to be identified for emission control and pollution management[24–26].

Here, we conducted field investigations of 13 industries, covering 122 factories. These industries are organic chemical production (OC), co-processing in cement kilns (CK), municipal solid waste incineration (MSWI), hazard waste incineration (HWI), coke production (COP), electric arc furnace for steel-making (EAF), iron ore sintering (IOS), primary copper smelting (PCu), secondary aluminum smelting (SAl), secondary copper smelting (SCu), secondary lead smelting (SPb), secondary zinc smelting (SZn) and coal fire power plants (CFP). Industrial fine particulate matter (i-PM) and chemical bottom liquids were analyzed to identify the 11 PHCZ congeners frequently detected in the environment (Supplementary Table 1). PHCZ congener profiles

of industrial samples were compared with those of environmental samples (Supplementary Fig. 2). Annual global PHCZ emissions from studied industrial activities producing PHCZs at relatively high concentrations were also estimated and mapped using the emission factor method.

## Results and discussion

### PHCZ concentrations in fine particle matter from multiple industries

Material samples potentially with PHCZs originated in the following industries: OC, CK, MSWI, HWI, COP, EAF, IOS, PCu, SAl, SCu, and SZn. Data obtained are shown in Fig. 1. The concentration of all PHCZ congeners (Σ11PHCZs) in an individual i-PM matrix and the mean Σ11PHCZs in i-PM matrices collections from all factories in each industry (removing minimax through box plot) were both used to describe the emission potential of each industry. The highest Σ11PHCZs in all i-PM samples (188 ng/g) and the highest mean Σ11PHCZs (30 ng/g) are associated with COP, in which matrices from COP-3, COP-5, and COP-1 had high contents: 85 ng/g, 59 ng/g, and 49 ng/g, respectively. Carbazole is an important by-product of coke production from coal tar[27]. The possible halogenation of carbazole might lead to highly elevated Σ11PHCZs in i-PM from coke production, making COP a significant industrial source of PHCZs. Relatively high Σ11PHCZs were found in i-PM samples from IOS, SAl, SCu, SZn, MSWI, and HWI, ranging from 8 ng/g to 60 ng/g. The IOS and SCu are

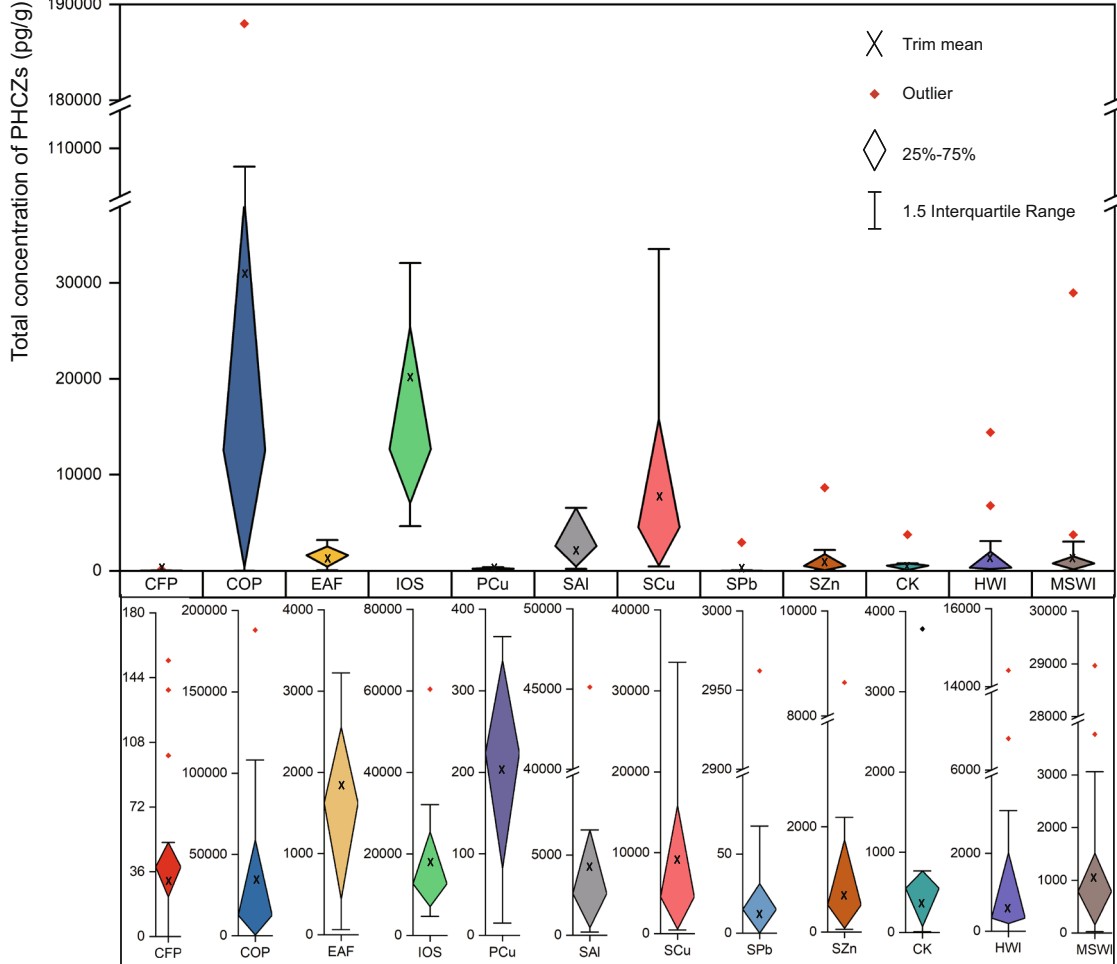

**Fig. 1 | The total concentrations of polyhalogenated carbazoles (PHCZs) in industrial particulate matter (i-PM) from 12 investigated industries.** CK, co-processing in cement kilns; MSWI municipal solid waste incineration, HWI hazard waste incineration, COP coke production, EAF electric arc furnace for steel-making; IOS iron ore sintering, PCu, primary copper smelting; SAl, secondary aluminum smelting; SCu, secondary copper smelting; SPb, secondary lead smelting; SZn, secondary zinc smelting; CFP coal fire power plants.

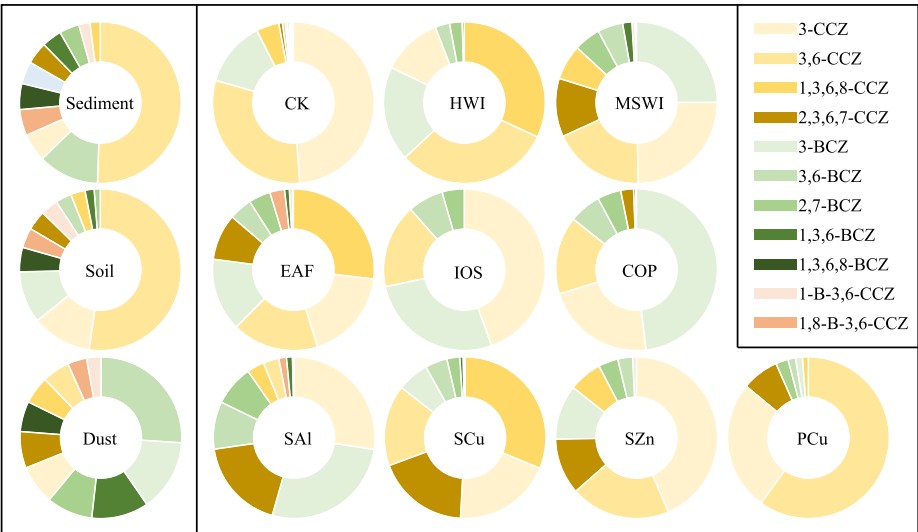

**Fig. 2 | Congener profile of polyhalogenated carbazoles (PHCZs) in the environmental medium and industrial particulate matter (i-PM) from industrial sources.** CK, co-processing in cement kilns; MSWI municipal solid waste incineration, HWI hazard waste incineration, COP coke production; EAF electric arc furnace for steel-making; IOS iron ore sintering, PCu, primary copper smelting; SAl, secondary aluminum smelting; SCu, secondary copper smelting; SZn, secondary zinc smelting.

considered high-level emitters, with mean Σ11PHCZs of nearly 20 ng/g and 10 ng/g, respectively, and at least two i-PM samples containing more than 15 ng/g PHCZs. The coke powder and cable cover of copper scrap (Supplementary Table 2) might be the major reason for the high PHCZ contents of i-PM from the IOS and SCu sources. Within each of MSWI, HWI, and SAl, Σ11PHCZs vary (Fig. 1) owing to discrepancies in raw material compositions[28]. Nevertheless, SAl-1, MSWI-3, and HWI-4 release i-PM with Σ11PHCZs higher than 14 ng/g, indicating their PHCZ emission potential. Some i-PM samples from CK, EAF, and SZn contain more than 3 ng/g PHCZs. Among investigated industries, coke-participating industries are significant but previously ignored PHCZ sources. Secondary smelting is also an important source of PHCZ emissions. In particular, SAl-1 emits i-PM with Σ11PHCZs as high as 45 ng/g and SCu-7 emits i-PM with Σ11PHCZs of 34 ng/g, which show their PHCZ pollution potential. However, Σ11PHCZs less than 68 pg/g can be seen in i-PM from SPb, except SPb-9, which is much lower than those from SAl, SCu, and SZn. This is consistent with the previously reported discrepancy in diverse secondary smelting sources of unintentional emissions of polychlorinated naphthalenes, which share similar mechanisms to the unintentional emissions of persistent organic pollutants (POPs)[29]. The limited ability of lead to promote the formation of POPs might be the reason for the relatively low Σ11PHCZs of i-PM from SPb[30]. Representing the waste disposal industry, MSWI releases i-PM with a greater PHCZ content than CK and HWI, for example, 29 ng/g PHCZs in i-PM from MSWI-3. The variety of raw materials and burning processes contribute to the differences, similar to variations observed for PCDD/PCDF released from waste incinerators[31]. For example, Σ11PHCZs from CK-1 are much higher than those from other factories (Supplementary Table 2), which is similar to results obtained for PCDFs reported in our previous study[32]. The incineration of laboratory wastes, such as plastic products and organic reagents, leads to higher Σ11PHCZs in i-PM from CK-1 compared with that in other i-PM from the incineration of industrial wastes (Supplementary Table 2). Apart from these, Σ11PHCZs as low as 0.37 ng/g are also observed in primary copper production, much lower than that in secondary copper production. Compared with the scrap smelted in SCu, the primary copper in PCu is relatively clean with a lower chlorine content[33], which inhibits the formation of PHCZs. Low Σ11PHCZs have been detected in i-PM samples from 15 CFP facilities owing to the possible inhibitory effect of sulfur in coal[34]. High Σ11PHCZs have been

detected in matrices collected from chemical manufacturing plants. Previous studies have demonstrated that the nitrobiphenyl group is an important precursor of bromocarbazole synthesis[35,36]. The existence of nitrobenzene, biphenyl, and chlorobenzene in material from chemical manufacturing plants was confirmed using high-resolution Q-TOF mass spectrometry screening (Supplementary Table 3). The co-existence of multiple precursors might lead to the formation of significant amounts of PHCZs during chemical manufacturing processes. Even though the bottom liquid requires further disposal, the high Σ11PHCZs from OC-1 (0.10 mg/L) and OC-2 (0.19 mg/L) demonstrate that certain OC processes are a significant industrial source of PHCZs.

The toxic equivalent PHCZ concentration (TEQ$_c$) of PHCZs in i-PM from SCu-7 is 15 pg/g and that of PHCZs in i-PM from SAl-1 is 8 pg/g, which greatly exceeds the TEQ$_c$ of PHCZs in other i-PM samples (Supplementary Table 2). Thus, these mentioned above industries seem to pose the highest PHCZ-related human health risk from PHCZs. Compared with i-PM from COP, i-PM products from SCu-7 and SAl-1 have lower PHCZ concentrations but higher contents of more toxic congeners, 1,3,6,8-CCZ and 2,3,6,7-CCZ, which leads to a higher TEQ$_c$ (Supplementary Table 4). Because PHCZs have a lower TEQ$_c$ than PCDD/Fs[31], PHCZs are not a major TEQ$_c$ issue in thermal processes.

## Congener profiles and homologue patterns of PHCZs in matrices from industrial activities

All target congeners were detected in i-PM material from COP, SAl, EAF, and MSWI (Supplementary Table 4), while specific congeners, especially chlorobromocarbazoles, were not detected in most i-PM from other industries. The proportions of congeners substituted with different halogens vary depending on the type of industry. As shown in Fig. 2, compared with other halogen-substituted carbazoles, chlorinated carbazoles count for more than 70% in samples from EAF, HWI, PCu, SCu, and SZn, demonstrating that chlorinated carbazoles are the dominant congener type in the unintentional production of PHCZs (data processing in Supplementary Method 1). Notably, 3-CCZ and 3,6-CCZ are dominant congeners, with total proportions ranging from 30 to 86% in all investigated industries. The proportion of 1,3,6,8-CCZ is high, exceeding 25% in SCu, EAF, and HWI, and although the proportion of 2,3,6,7-CCZ is only 18% in SCu, it is the highest in the other industries. Brominated carbazoles were also detected in all i-PM samples, although high-bromine congeners, especially 1,3,6,8-BCZ, had a

very low detection frequency and were found in only several i-PM samples from CK, MSWI, and EAF. 3-BCZ is the major brominated congener, especially in COP (48%), SAl (27%), and IOS (27%). The proportions of 2,7-BCZ (8%) and 3,6-BCZ (10%) in SAl are much higher than those in other secondary metal smelting processes. The two bromo-substituted carbazoles might be particular to PHCZ production in SAl, distinguishing SAl from other secondary metal smelting processes. Only three industries (EAF, SZn, and MSWI) seem potential producers of chlorobromocarbazoles, with the highest proportion of only 4% in EAF. The proportion of chlorinated carbazoles is higher in OC than in other industries owing to the presence of certain raw materials and the specific synthesis pathway. Specifically, the chlorine content is considerably higher than the bromine content, which is also favorable for the formation of halogenated dibenzo-*p*-dioxins and dibenzofurans[37,38].

## Characteristics of PHCZs from industrial sources

Spearman analysis of the congener concentrations and environmental occurrences of PHCZs produced by the investigated industries (see Supplementary Table 4 and Supplementary Fig. 2) was conducted to understand the source and environmental characteristics of PHCZs (data processing in Supplementary Method 1). As shown in Fig. 3 (brown and green indicate positive and negative correlations, respectively), certain congeners display a strong positive correlation with other congeners in SAl, MSWI, and CK. In the other industries, such as COP, SZn, and IOS, correlations exist between minority of congeners including 3-CCZ, 3-BCZ, 3,6-CCZ, and 1,3,6,8-CCZ. Interestingly, strong negative correlations between certain congeners in the environment are absent in every industry. For example, 3,6-CCZ has negative correlations with 10 other congeners in sediment and soil but weak or positive correlations with these congeners in all industries. The proportion of 3,6-CCZ in i-PM samples is high but the correlations between 3,6-CCZ and other congeners in the environment and industries differ, suggesting that anthropogenic and natural sources might jointly contribute to 3,6-CCZ pollution in sediment. The result is consistent with findings reported by Guo et al.[14], but more field studies are needed to further confirm pollution contribution from these investigated sources as well as natural sources and other unexplored artificial sources in specific areas. The mutual interferences of 3,6-CCZ and pesticide DDT[39,40] in laboratory analysis should also be taken into consideration when the discrepancy appeared. The concentrations of 1-B-3,6-CCZ, 1,8-B-3,6-CCZ, 1,3,6-BCZ, 1,3,6,8-BCZ, and 2,3,6,7-CCZ are low, while the concentrations of 2,7-BCZ and 3,6-BCZ are high in some i-PM samples, such as from COP-1 and SAl-1 (Supplementary Table 4). 3-CCZ, 3-BCZ, 3,6-CCZ, and 1,3,6,8-CCZ exist in high concentrations in i-PM samples from most industries, indicating that industrial activities are part of their sources.

The Σ11PHCZs in i-PM samples are high and consistent with environmental investigations, suggesting that industrial activities are sources of some congeners. However, the diverse correlations among congener concentrations in industries and the environment indicate the existence and important contribution of natural sources.

## Comparison of congener and homologue patterns between i-PM and environmental samples

The proportion of chlorocarbazoles, ranging from 62–94%, is higher than those of bromocarbazoles and chlorobromocarbazoles in most industries (excluding CFP and SPb because of the much lower Σ11PHCZs). The proportion of brominated carbazoles is high in i-PM samples from COP (56%) and SAl (45%) but ranges from 4 to 38% in i-PM samples from the other investigated industries. The proportion of chlorobromocarbazole is 0–4% of Σ11PHCZs. These results coincide with the concentrations in sediment (63% chlorocarbazoles, 30% bromocarbazoles, and 7% chlorobromocarbazole) and soil (71% chlorocarbazoles, 21% bromocarbazoles, and 8% chlorobromocarbazole)

(Fig. 2). Specifically, 3,6-CCZ is the dominant congener, constituting nearly 50% of Σ11PHCZs in both soil and sediment. 3-CCZ and 3-BCZ are also important congeners, with proportions of 13 and 9% in soil, while 3,6-BCZ (12%) is an important congener in sediment. In our research, 3,6-CCZ is considered the main congener and constitutes 15–60% of Σ11PHCZs in industries other than SAl. 3-CCZ is the dominant congener owing to its high proportion (18–45%), which is slightly higher than that of 3-BCZ (1–48%). Wu et al.[5] reported that 3,6-CCZ accounts for more than 70% of Σ11PHCZs in sediment from San Francisco Bay. A similar congener pattern was obtained for samples from China. Sediment and soil samples from Jiangsu[4], Zhejiang[39], and Yunnan provinces[41] all contain 3,6-CCZ with proportions from 55 to 86%. In some early studies, 3-CCZ and 3,6-CCZ were detectable while other congeners are not. The concentration of 3,6-CCZ is up to 50 ng/g in Lipple River[42] and 3500 ng/g in industrial coastal areas of Greece[2]. The concentrations of 3-CCZ and 3,6-CCZ in soil from Bavaria were 8.3 ng/g and 149 ng/g, respectively[43]. The preponderance of 3-CCZ, 3-BCZ, and 3,6-CCZ in both i-PM and environmental samples indicates that PHCZ pollution is significantly influenced by industrial sources.

Principal component analysis of PHCZ congener profiles of i-PM samples and environmental occurrences was also conducted (Supplementary Method 1). As shown in Fig. 4, almost all environmental occurrences are classified into one group, in which some i-PM samples from CK, HWI, MSWI, SZn, and SCu are also grouped. Noticeably, i-PM samples with high PHCZ concentrations, including CK-1, HWI-3, SCu-6, and MSWI-3, are also found in this group. Most sampling sites of environmental investigations of PHCZs are in or surrounded by big cities (Supplementary Fig. 2), such as San Francisco in the USA and Wuxi and Dali in China, which generate considerable amounts of municipal solid wastes and scrap non-ferrous metals. Wu[4,5] pointed out that although the sources of PHCZs in San Francisco Bay and Lake Tai have not been determined, the abundance of PHCZs indicates that anthropogenic activities are likely sources. The results show that secondary metal smelting and waste incineration, which are intense in populated areas, might be potential sources of PHCZ residues.

Historically, there used to be many industrial activities around Lake Tai and other PHCZ-polluted sites. However, without PHCZ emission data, these anthropogenic sources have not been identified. The present research provides not only data for the potential industrial sources of PHCZs in these areas but also a set of PHCZ fingerprints from long-neglected thermal sources, triggering further work for the establishment of a PHCZ fingerprint database.

## Preliminary estimation of PHCZ release from priority industrial sources into the environment

Although PHCZs are generally released from the 13 investigated industries, no global emission estimate could be allocated to activities producing i-PM with a low PHCZ content, such as PCu, secondary lead smelting (SPb), coal-fired power (CFP), CK, HWD, and SZn (Section 3.1). An estimation of PHCZ emission from OC with a high concentration in the first two processes investigated (tetrachloroethylene and chlorobenzene production) is not given either because further comprehensive investigations are required. However, our field study of Yaer Lake (Supplementary Fig. 3) demonstrated the significance of historical chlor-alkali industries (via gas diffusion, solid residues, and wastewater) for environmental PHCZ residues. The rest of the investigated industries, namely high-polluting industries including EAF, SAl, COP, SCu, SZn, and MSWI, are given an estimation of PHCZ emission.

The biggest emitter among the investigated industries is COP, contributing 9229 g of PHCZs through the release of industrial fine particles in total per year globally, representing a high contribution. In 2019, the global production of coke approached 683 million tonnes (Fig. 5a), the highest among these industries[44]. Moreover, COP consists of several procedures that emit particulate matter (PM)[45] (Supplementary Table 5), such as coal loading, coke discharging, coke oven

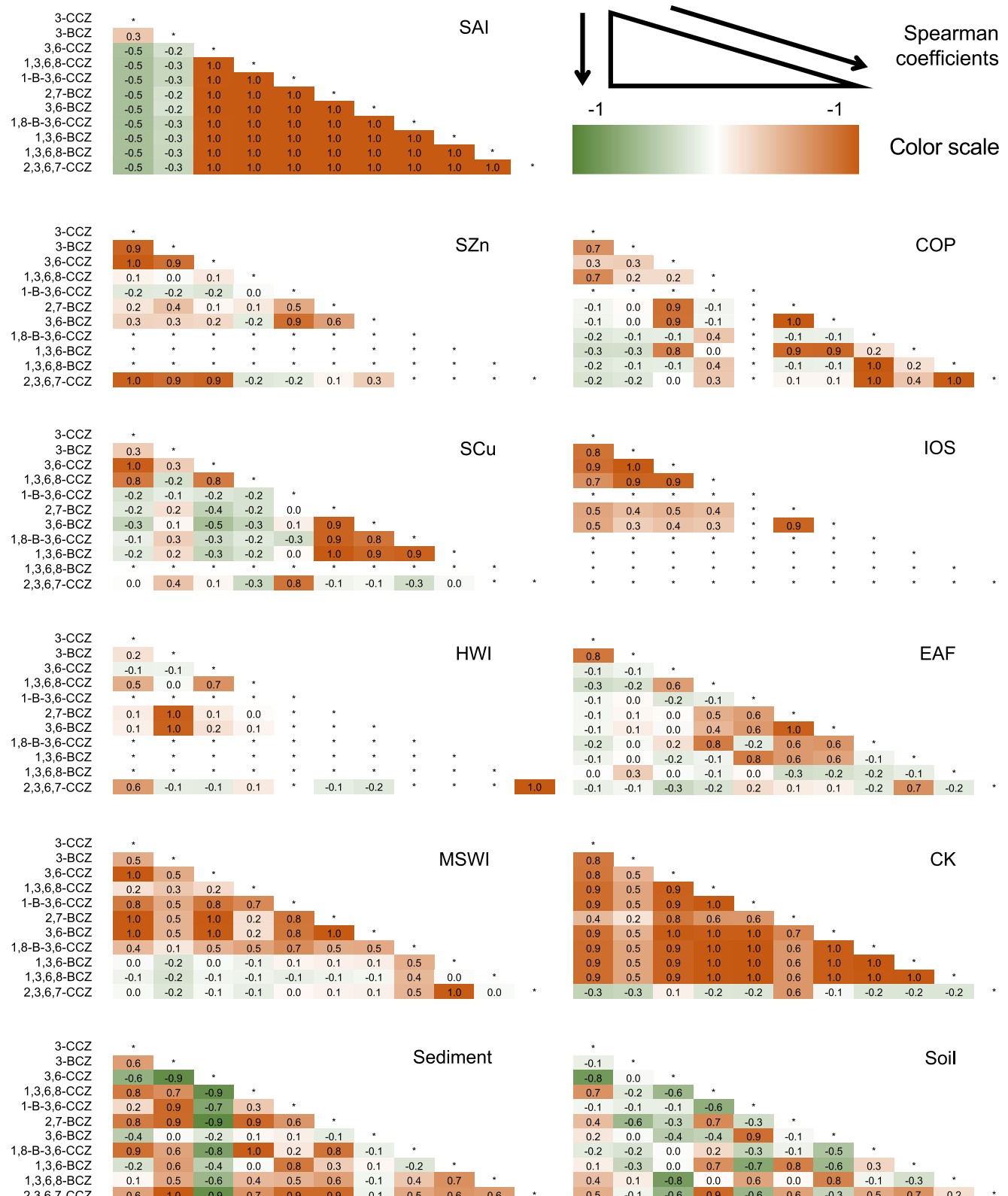

**Fig. 3 | Spearman's coefficient of congener concentrations of polyhalogenated carbazoles (PHCZs) in sediment, soil, and industrial particulate matter (i-PM).** CK, co-processing in cement kilns; MSWI municipal solid waste incineration, HWI hazard waste incineration, COP coke production, EAF, electric arc furnace for steel-making; IOS iron ore sintering, SAI, secondary aluminum smelting; SCu, secondary copper smelting; SZn, secondary zinc smelting.

heating, and dry coke quenching. The total emission factor of i-PM (EF) is relatively large, close to that of secondary metal smelting (Fig. 5b). The high yield, EF[45], and PHCZ concentration make COP the primary source of PHCZs (Fig. 5c). China is one of the important contributors of

PHCZ global emissions, releasing 6364 g of PHCZs every year (Fig. 6) due to the high industrial activities. The other regions, including Russia, India, America, Japan, and the European Union, also emit significant amounts of PHCZs, ranging from 149 g to 472 g. China was the

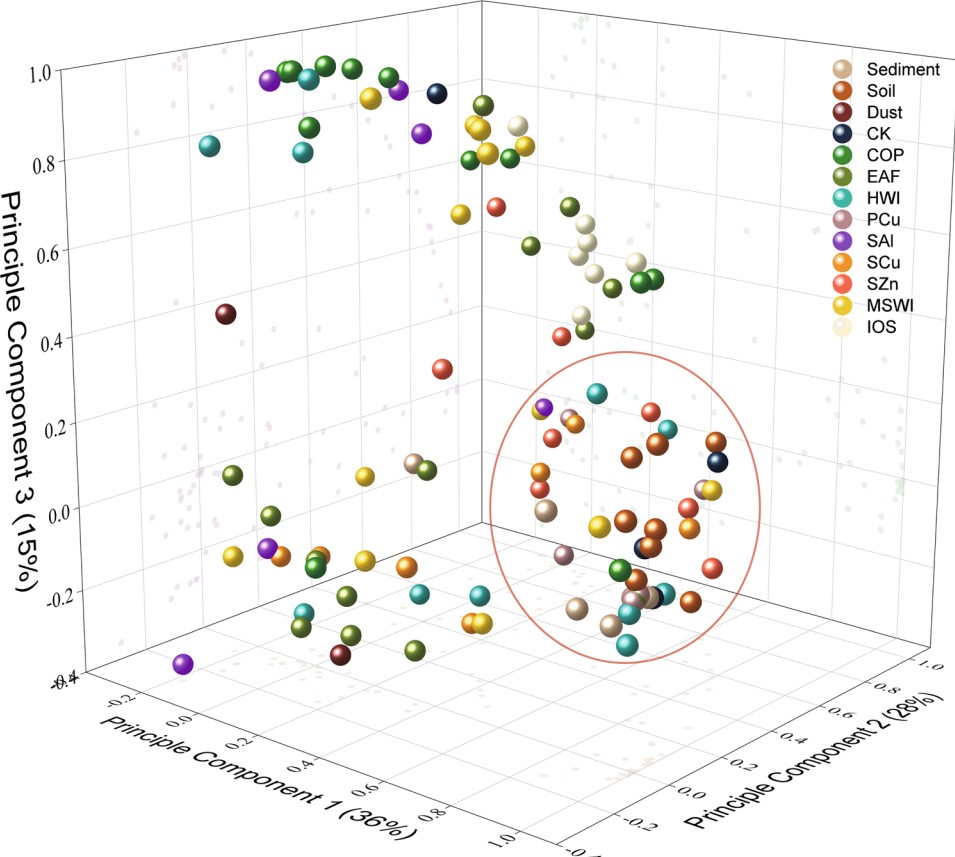

**Fig. 4 | Clustering of the environmental occurrences and industrial particulate matter (i-PM) from industrial sources.** CK, co-processing in cement kilns; MSWI, municipal solid waste incineration;HWI: hazard waste incineration;COP coke production, EAF, electric arc furnace for steel-making, IOS iron ore sintering; PCu, primary copper smelting; SAl, secondary aluminum smelting; SCu, secondary copper smelting; SZn, secondary zinc smelting.

biggest coke producer in the world in 2019, producing 471 million tonnes (Supplementary Fig. 4), which explains its high PHCZ emission. IOS is another significant PHCZ source and can contribute 3237 g of PHCZs into the environment every year, the second highest emission of PHCZs. Australia has been considered as the biggest emitter of PHCZs, accounting for 1/3 of the global annual emission of PHCZs owing to its high production of iron ore (Supplementary Fig. 4), while Brazil, United States, India, and China annually emit more than 300 g of PHCZs.

EAF has relatively high global emissions of PHCZs among the high-polluting industries. The annual PHCZ emission of 90 g is attributed to EAF, which is higher than that from SCu (Fig. 5c). EAF has similar industrial outputs to COP, approaching 523 million tonnes[46] in 2019 (Fig. 5a), resulting in considerable PHCZ emissions despite the lower Σ11PHCZs and EF of PM[45]. SCu emits 18 g of 11 PHCZs into the environment every year. The low PM (Fig. 5b) discharge reduces its PHCZ input into the environment. Mapping of PHCZ emissions from EAF (Fig. 6) indicated that China, America, and India are the top three emitters of PHCZs, with emissions of 19 g, 11 g, and 11 g, respectively. Other countries, such as Japan, South Korea, Iran, and Turkey, potentially contribute to the relatively high PHCZ discharge of 4 g per year through EAF, in contrast to COP (Supplementary Fig. 4). MSWI and SAl are not major emitters, with global PHCZ emissions of less than 7 g per year (Fig. 5c) owing to low production outputs[47,48] (Fig. 5a) or EF of PM[49] (Fig. 5b). Therefore, pollution control is a priority for COP and IOS and less urgent for EAF and other industries.

These estimated industrial sources might deliver a more considerable amount of PHCZs into the environment in historical industrial activities than now, considering that POP reduction technology had not been intensively adopted historically. The mapping of these results showed that PHCZs are a global environmental pollution issue and a human health risk. As for source emission assessment, further research should be conducted for (1) emissions from these industries in other countries to develop more robust emission factors for different technology levels; (2) investigation of PHCZ emissions from other more industrial processes; (3) field studies or model-based to quantify PHCZ emissions from historical industrial activities; and (4) comprehensive investigations of natural PHCZ source emission to clarify their significance for PHCZ pollution.

## Methods
### Sample information
In total, 122 industrial plants from 13 industries were included in this study. This comprehensive field investigation enables source identification and emission estimation of POPs in one single study[31]. i-PM samples were collected using bag filters, the most popular and effective dust-removing device in the world today. Each i-PM contained a mixture of PM generated during at least 48 h of regular production. The bag filter was placed in the total discharge outlet and represented the PHCZ release level. The particle size of each i-PM sample was normally lower than 2.5 μm (Supplementary Fig. 5). Airborne PM was formed when the sample was released into the atmosphere.

Specifically, the i-PM samples were from 15 CFP plants, 6 CK plants, 13 MSWI plants, 12 HWI plants, 13 COP plants, 14 EAF plants, 7 PCu plants, 7 SCu plants, 6 SAl plants, 8 SZn plants, 11 SPb plants, and 8 IOS plants. Chemical bottom liquid samples, collected from the reaction still in the final manufacturing step of the end product, were from 2 OC plants producing tetrachloroethylene and chlorobenzene.

Some research suggests that the load, mainly the PM, is the original site for the formation of numerous heterogeneous organic

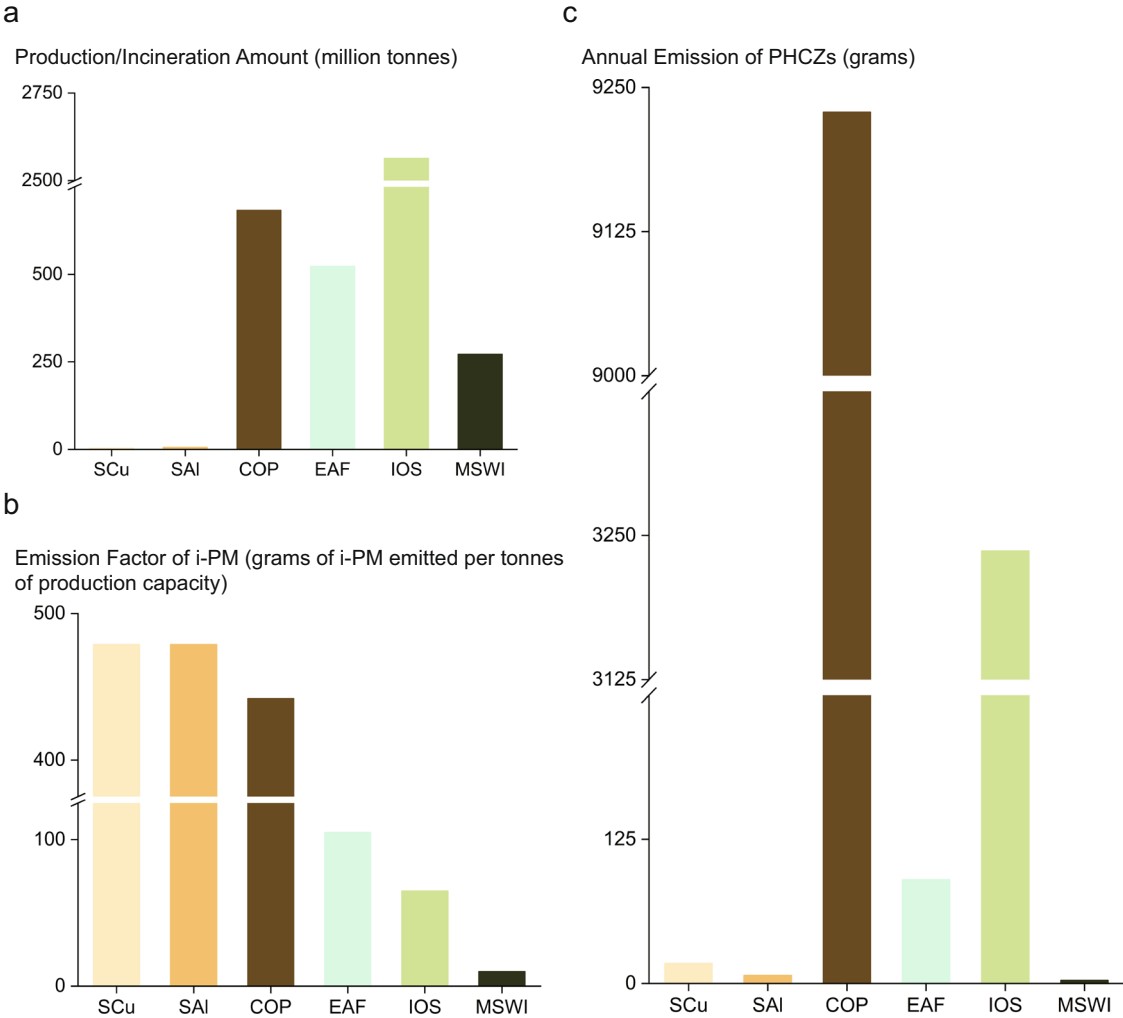

**Fig. 5 | Information on the industrial activities and estimated emissions for selected industries. a** Production/incineration amount[44, 46–48,55], **b** emission factors[45,49] of industrial particulate matter (i-PM) and **c** estimated annual global emission of polyhalogenated carbazoles (PHCZs) of selected industries. MSWI, municipal solid waste incineration; COP, coke production; EAF, electric arc furnace for steel-making; IOS iron ore sintering, SAl, secondary aluminum smelting; SCu, secondary copper smelting.

pollutants, which contribute to the environmental input[50]. The Σ11PHCZs of i-PM samples are accepted to represent the industrial emission situation of PHCZs.

## Chemicals and reagents

The standard mixture of 11 PHCZs, namely 3-chlorocarbazole (3-CCZ), 3,6-CCZ, 1,3,6,8-tetrachlorocarbazole (1,3,6,8-CCZ), 2,3,6,7-CCZ, 3-monobromocarbazole (3-BCZ), 2,7-dibromocarbazole (2,7-BCZ), 3,6-BCZ, 1-bromo-3,6-dichlorocarbazole (1-B-3,6-CCZ), 1,8-dibromo-3,6-dichlorocarbazole (1,8-B-3,6-CCZ), 1,3,6-tribromocarbazole (1,3,6-BCZ), and 1,3,6,8-tetrabromocarbazole (1,3,6,8-BCZ), with a purity of >98% was purchased from Wellington Laboratories (Ontario, Canada). [13]C-1,3,6,8-marked CCZ ([13]C-1,3,6,8-MCCZ) and [13]C-3,6-marked CCZ ([13]C-3,6-MCCZ) (2.5 μg/mL, Wellington Laboratories) were all diluted to 1 μg/mL with n-hexane and used as a cleanup standard and injection standard, respectively. Anhydrous sodium sulfate was baked at 600 °C for 6.5 h. Silica gel was activated at 450 °C for 6.5 h.

## Sample pretreatment and instrumental analysis

The detailed analysis method is described in Supplementary Method 2. Briefly, 10 ng of [13]C-1,3,6,8-MCCZ as the surrogate standard was added to 5 g of i-PM. After 2 days of aging, the mixture was Soxhlet extracted for 12 h using acetone/n-hexane (1:1, v/v). Then, the extract was

purified using a glass column packed with silica and anhydrous granulated sodium sulfate. After concentration, 10 ng of [13]C-3,6-MCCZ was added as the injection standard to the eluate, which was analyzed using a gas chromatography (GC) system coupled to a triple quadrupole mass spectrometer (MS/MS) to achieve target compound detection. Chemical bottom liquid samples were similarly purified before GC–MS/MS analysis.

## Quality control

The method has been evaluated in terms of method stability and sensitivity in our previous research. The recoveries of all congeners in parallel spiked samples range from 82 to 137%, and the relative standard deviations (RSDs) range from 2.4% to 35.8%. The method detection limit of each congener ranges from 1.46 to 3.82 ng/mL for liquid samples and from 0.009 to 0.023 ng/g for solid samples. In this research, 11 PHCZs were not detected in the blank control samples, and the recoveries of surrogate standards were in the range of 54–118%.

## Emission and TEQ$_c$ calculation

Quantification of POP release has been formulated as a set of systematic calculations method on the basis of numerous research[51,52], which is documented in open-source guides, such as the Toolkit for PCDD/Fs published and endorsed by UNEP and the Stockholm

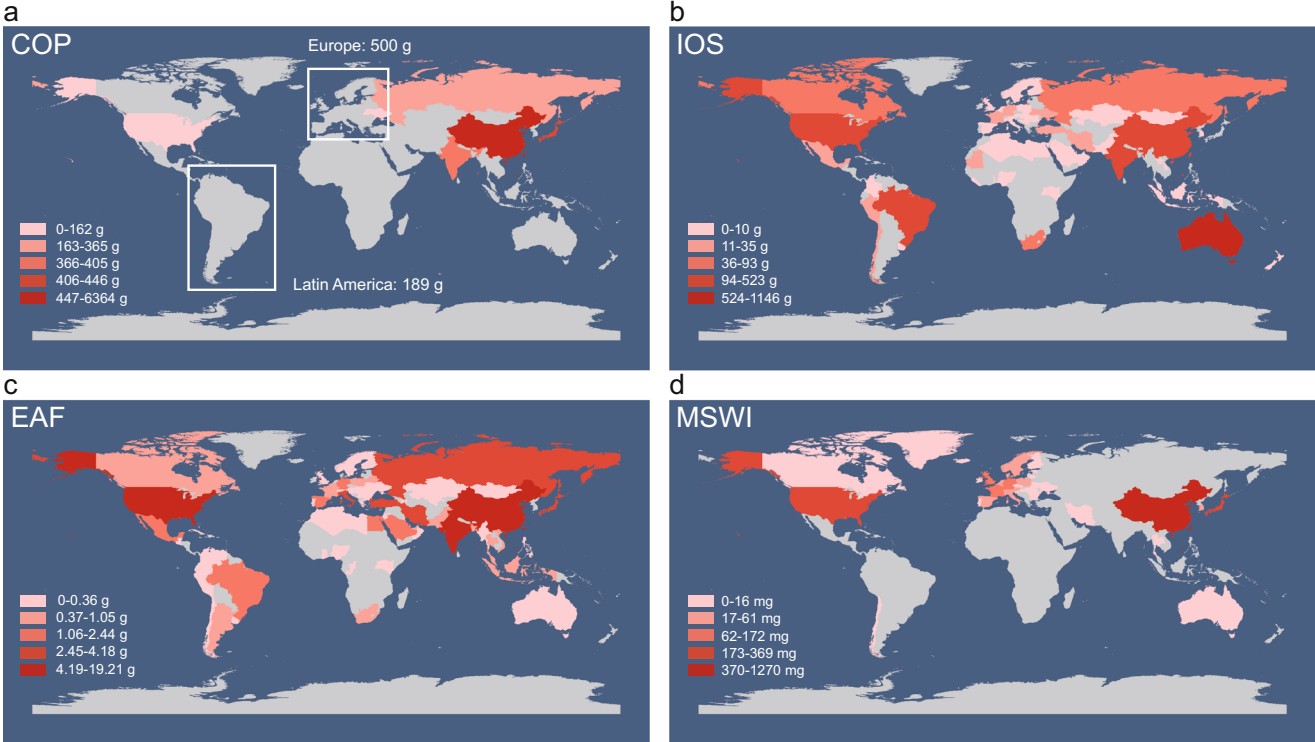

**Fig. 6 | The preliminary estimation of annual global emission of poly-halogenated carbazoles (PHCZs) (grams/milligrams per year) through industrial particulate matter. a** coke production (COP), **b** iron ore sintering (IOS), **c** electric arc furnace for steel-making (EAF) and **d** municipal solid waste incineration (MSWI) based on 2019 production data. Map sourced from Database of Global Administrative Areas free vector data via ArcGIS/Esri.

Convention[53]. Computational models are common in emission estimation[54], largely owing to the scarcity of analytical methods and samples, while emission factors are based on experimental data directly representing real emissions[31]. To estimate the emissions of 11 PHCZs in high-polluting industries as well as some countries and regions, we adopted equation (2):

$$E = \bar{C}_i \times EF_i \times Y_i \tag{1}$$

where $E$ represents the annual emission of PHCZs in each high-polluting industry, $\bar{C}_i$ is the average value of Σ11PHCZs in i-PMs from all factories (removing minimax through box plot) in the same industry, $EF_i$ represents the average emission factor of fine particulate matter (PM) (described in Uncertainty Reduction), and $Y_i$ is the yield in a specific industry. $Y_i$ (Fig. 5b) is the annual production of each industry around the world, which was collected from relevant international statistics organizations[44,46–48,55]. $EF_i$ (Fig. 5c) was collected from China's Industrial Source Pollution Production and Discharge Coefficient Manual and research results[45,49]. The yields of SCu and SAl are for 2021, and the yields of COP, EAF, IOS, and MSWI are for 2019. We also collected the yields of EAF (Supplementary Fig. 4), IOS (Supplementary Fig. 4), COP (Supplementary Fig. 4), and MSWI (Supplementary Fig. 4) in major regions, which was applied for mapping of the regional emission of PHCZs (Fig. 6). Because of the lack of usable data for SCu and SAl, only the global emissions are given in Fig. 5c.

Details for calculating the TEQ_c and relative effect potencies (REPs) of PHCZs can be found in Supplementary Method 3 and Supplementary Table 6, respectively.

## Uncertainty

Uncertainty is unavoidable in estimations based on the emission factor owing to variations of production procedures and facilities, measuring time, pollution control measures, and diverse raw materials[56,57]. Here, some measures were taken to reduce uncertainty. For example, i-PM samples were collected in bag filters as PM emissions over several days owing to the constant ash-dumping period, making the PHCZ concentration a steady emission situation. More than six industrial plants were sampled for each industry (Supplementary Table 2). To reduce the variation associated with diverse plants, the box plot was used to exclude the minimax of Σ11PHCZs in each industry, and then the average Σ11PHCZs of diverse factories was used. Moreover, the EFs of SCu, IOS, COP, and EAF were determined following a nationwide survey of pollution emissions with the help of each industrial association in China, having been the recommended handbook for the Second National Pollution Source Census, indicating their reliability.

The EF of MSWI is the value in latest research[49], and it is close to the values in other regions. To our best knowledge, a reliable EF is unavailable for SAl. Pyro smelting is mainly used to smelt scrap aluminum, which is also used in SCu. Therefore, the EF of SCu is used to estimate the PM emission of SAl. As shown in Supplementary Table 5, the EF of PM is determined by various factors in various industries. For COP, the sum of the EFs of PM released from four procedures is the final EF of PM. Wet coke quenching is not considered here because of its low usage rate. For example, in China, wet coke quenching is only allowed to operate when the dry coke quenching machine is under maintenance. The final EF of PM is the average of the values of two production technologies (top-charging and stamp-charging). For EAF, the end product is the core factor affecting PM emission. Carbon steel accounts for 80% of steel production, which is the reason the EF of carbon steel production represents the overall emission level. For IOS, the EF of PM is determined by sintering machines, of which more than 360 m² is used because of the popularization of large-size machines. The raw materials in SCu production determine the EF of PM. The average of EFs determined by high-grade scrap copper and low-grade scrap copper is used here for comprehensive calculation.

## Data availability

The minimum dataset necessary to interpret, verify and extend the work is provided in the supplementary information. The concentrations of ployhaleganated carbazoles for Figs. 1–6 are provided in supplementary information. The published data about congener concentrations and profiles in the environment used in Figs. 2–4 is from references summarized in supplementary Fig. 2. Data in Figs. 5 and 6, Supplementary Fig. 4 and Table 5 for emission assessment is collected from research results, international industry associations and official government document, which are introduced in Method and provided in references. Unpublished data is used for supplementary Fig. 3.

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

## Acknowledgements

This work was supported by the Second Tibetan Plateau Scientific Expedition and Research Program (STEP) (grant number 2019 QZKK0605), the National Natural Science Foundation of China (grant numbers 92143201, 22076201, 21936007 and 21906165), and the CAS Interdisciplinary Innovation Team (grant number JCTD-2019-03).

## Author contributions

Y.S. conducted the laboratory analysis of samples, data analysis, and wrote the paper; L.Y. and M.Z. conducted part of the data analysis and interpretation; R.W., F.J., and L.G. conducted the analysis, evaluation, and interpretation of data and supported manuscript writing; C.Z. conducted part of the laboratory analysis of samples; C.C. and Q.Y. collected relevant information; G.L. designed the research, conducted data analysis, and revised the paper.

## Competing interests

The authors declare no competing interests.
