## [Peer Review File - NEW · Nature Communications]

Industrial source identification of polyhalogenated carbazoles and preliminary assessment of their global emissionsReviewer #1

Comments on manuscript (ID: NCOMMS-23-01489):

Polyhalogenated carbazoles (PHCZs) are structural similar to that of polychlorinated dibenzofurans (PCDFs) that attracting global concerns on their sources and control. The toxicity of PHCZs is like PCDFs with high carcinogenicity. However, compared to PCDFs, very little attention was given to the potential sources of PHCZs. The authors conducted many monitoring activities of polyhalogenated carbazoles in particle matters collected from a big size of industrial plants on real full-scale. This study provides essential new data for recognizing important sources of PHCZs and implementing their controlling strategy. I recommend minor revisions before it could be accepted.

Specific comments:

- (1) I suggest adding a figure in the supplementary information to display the molecular structure of polyhalogenated carbazoles with numbering of carbon atoms, as well as the structures of the detected congeners in the study. It could help to easily understand their congener profiles.
- (2) The authors compared the concentrations of PHCZs in different industrial sources and make reasonable explanations for most of the studied industrial sources. However, the low concentrations of PHCZs in lead smelting and coal fired power plants were not explained. The authors should discuss the possible reasons of the low concentrations. In addition, the authors should explain why high concentrations of PHCZs in PM from coke production?
- (3) What are the potential release pathways for organic chemical plants? Is it possible for PHCZs to enter environment by waste water, gas diffusion, solid or liquid residues like those of dioxin emissions? Please add relevant discussion.
- (4) Are the process techniques adopted in the plants the dominant in each of the investigated industrial categories in China or in the world?
- (5) In the Supplementary Information 1, relative effect potencies (REPs) of PHCZs based on the structure-dependent induction of cytochrome P450 1A1 mRNAs in MDA-MB-468 breast cancer cells were not listed. Please list those REPs of PHCZ congeners that adopted for TEQ calculations.
- (6) Supplementary Information 3. The information in this section, such as Production Equipment/Scale, Raw Materials should be checked and confirmed. Description about Coke Oven Chamber in the Table seems to be incorrect.

Reviewer #2

This is a timely report on the investigation of anthropogenic source of polyhalogenated carbazoles (PHCZs), a new class of emerging contaminants with considerable environmental loads and risk. Although increasing studies have suggested that PHCZs in the environment could be attributable to anthropogenic and natural sources, the formation/emission pathways are still inclusive. This study for the first time investigated the emissions of many previously unrecognized sources of PHCZs from industrial activities. The emissions of PHCZs from 12 large-scale industries were identified and evaluated. The presented source emission data is very abundant and important for interpreting the widespread occurrence of PHCZs in various environments. I recommend to accept this manuscript for publication after minor revisions or explanations.

Specific comments:

1. Did the authors characterize the fine particle matters collected from the large-scale industrial sources? I suggest providing a few examples (such as 1-3 samples) to display the basic properties of the particle matters from industrial sources, such particle size.
2. "EAF (electric arc furnace steel)" should be changed to "EAF (electric arc furnace for steel-making)".
3. Eleven PHCZ congeners were detected. There are more congeners than detected in this manuscript. Could the authors further explain why those eleven congeners were selected in this study?
4. The authors calculated the toxic equivalents (TEQs) of PHCZ emissions. What are the toxic equivalent factors (TEFs) used for calculating the TEQs of PHCZs? Are those TEFs widely used? Are they similar to those of dioxin congeners with TEFs issued by WHO?
5. I suggest the author providing a figure about the congener profiles of PHCZs formed during production of indigo dyes and photoelectric materials in the supplementary information. It could

provide a visual comparison of different congener profiles from different industrial activities.

6. High concentrations of PHCZs in bottom residue liquid samples from organic chemical production were detected. However, the description about the information of production amounts of investigated chemicals on national scale or global scale is inadequate. I suggest supplementing basic information about the production amounts of investigated chemicals for better judging the potential importance of chemical production sources.

7. Could the authors provide a brief description about the temporal trend of PHCZ emissions from the important industrial sources? For example, recent five or ten years.

8. Line 65: Cl-/Br- instead of Cl/Br.

9. Line 107: Change "detection rate" to "detection frequency" and throughout.

10. Line 148-149: The formation of PCDFs from the coupling reaction of chlorophenol and chlorobenzene under high temperature has been well studied. However, it is unclear whether PHCZs can be produced via similar pathways. Could the authors expand more on the possible formation pathways of PHCZs from these industrial activities?

11. As shown in Figure 1, high concentrations of PHCZs are emitted from coke production, secondary aluminum smelting, and municipal solid waste incineration. However, the emitting PHCZ concentrations from the three activities varied greatly from different plants. As for the same industry, are these plants used the same raw material and process for production? Could the authors explain why? What are the concentrations used for the estimation of the global emissions of PHCZs from these industries, since the emitting concentration of PHCZs varied greatly with different plants.

12. What is data source of the PHCZs in the environment presented in Figure 2? My understanding is that the figure analyzing the correlation of PHCZ congeners between fly ash and environmental samples. How can these results be used to interpret the occurrence of PHCZs in the environment?

Reviewer #3

This paper, for the first time, reports the concentrations of PHCZs in particulate matter (and bottom liquid) generated from industries. A total of 26 samples were collected from 26 plants of eight categories of industry, and 11 PHCZ congeners were analyzed in each sample. Efforts were made to examine the congener pattern, explore correlations between industrial emissions and environmental occurrence, and estimate the global industrial release of PHCZs. Such work is clearly important because, as the authors mentioned in L82-84 and I agree, that "current research on source identification is woefully inadequate, and knowledge to date does not support any reasonable explanation of the huge residues, wide distribution and complicated congener profiles of PHCZs in the environment". To this end, the necessity of conducting this work is justified. I also consider the experimental dataset valuable and would like to encourage its publication in an environmental journal. However, for the ambitious goal of providing a global assessment, the amount of work is too small, the conclusions are not solid, and the overall findings are too weak. For this reason, I do not recommend the publication of work in Nature Communications.

Major comments

Natural production of PHCZs is not the focus of this paper, but its importance must not be downplayed. In this paper, natural generations of PHCZs from fungi and volcano are mentioned in two sentences (L64-68), ignoring potentially significant (some are currently unknown) others. Based on this work, the total global emission of PHCZs from the eight industries is only about 10 kg/yr. How does this emission estimate compare with that for dioxins from the same industries? Such a low emission, if correct, could mean that industrial emissions be insignificant compared with natural production, as far as total PHCZs concerned.

Where were all the samples collected? Is any from outside China? How are the samples (each from one filter bag used for 48 hr) representative even within China? Additionally, each industry was represented by only 1-4 samples. L461-462 mentions that 2 to 4 samples from each industry are "comparable to the open guidance published by some authorities", but these open guidelines are not provided. Within each industry, the reported concentrations differ by orders of magnitude. For example, global emissions of 8650 g PHCZs from coke production was estimated from only two COP samples which differ by 11 times in concentration (the first two rows in Table 1). Similarly,

the only three samples of MSWI ranged from 0.1 to 29 ng/g (Table 1). The averages were used in the estimations for global emissions in this paper. Such estimates bear huge uncertainties which diminish the differences among industries and make the conclusions in this paper (for example, "The biggest emitter among the investigated industries was COP" in L289) very doubtful.

The conclusion that "the congeners in the industrial emissions and environmental occurrences exhibit similar grouping" (L337-338) need more caution. The idea that positive correlations indicate cause-effect relationship is wrong. The section starting in L200 is simply a correlation analysis among congeners, it does not support the concluding statement "industrial activities are sources of PHCZs" (L216) "in the world" (L200). Similar comment can be made on the section starting in L218. Congener 1368-BCZ is believed to come from legacy dye industry, which was not involved in this work; thus L238-246 is not needed.

Throughout the paper, the concepts of congener and homologue are messed up. A total of 11 congeners were measured in this work, and they are in three homologues – chlorinated, brominated, and chlorobromo- carbazoles. The section starting in L162 has "congener" in the section title, but the content is mostly on homologues. In many places, the word "homologue" is used but individual congeners are focused.

The English is understandable, but lacks the concision and brevity. Some grammatical errors are present, especially towards the later section (e.g. Method). There are many redundant sentences and expressions in this paper. Significant efforts are needed to meet the criteria for technical writing, especially for publication in highly ranked journals.

Specific comments

L43: References 2-5 appear to be randomly selected. Randomly picking of references should be avoided. Suggest removing Refs 1-5 or moving them to specific places in the paper where they fit more specifically.

L40-53: Given the focus of this paper is on PHCZ sources, the entire first paragraph about toxicity can be deleted for brevity and to avoid misunderstanding on the focus of this paper.

L60: Two papers should be cited along with Ref 25: <https://dx.doi.org/10.1021/es503936u> and <https://doi.org/10.1021/acsestwater.2c00191>. Please also replace "concentrations of 38 ng/g led to" with "a median concentration of 23.7 ng/g in surface grabs, and". The number 23.7 can be found in Li et al. 2022 Table S4. This number does not "lead" to the total load of 3354 tonnes in Table S6.

L60-63: Delete these two sentences. Although nothing seems wrong, ambiguous words such as "huge" lower the quality of writing.

L82: Move the comma to after "inadequate".

L86-93: Suggest deleting these redundant statements.

L95: "particulate matter", not "matters" (no "s"). Please check throughout the paper.

L96: Suggest ending the sentence after "liquids" and starting a new sentence "The results were examined for links and correlations with reported congener profiles in environmental samples, with the aim of identifying sources". SI-2 and SI-4 should be mentioned in the Result section, not here.

L107-112: Suggest dividing this long sentence into two. One is "The industries included". Then, "Data obtained are shown in Fig. 1a".

L163-176: Suggest deleting these text "Emissions So," which sound redundant and help nothing.

L171-172: Suggest changing "some of them contained a large amount of PHCZs" to "the PM from

some of these industries had high concentrations of total PHCZs”.

L195: The starting sentence is incorrect. This study is NOT the first discovering industrial sources of PHCZs. This paragraph is no use and should be deleted for brevity.

[Specific comments above are mostly editorial and for Introduction only.]

Fig 1: SA1 < COP in concentration, but SA1 > COP in TEQ. The reason might be the higher content in SA1 of 2367-CCZ and 27-CCZ which are more toxic. If true, a brief explanation should be added to main text L153-155.

Fig 2: Please deleting half of the data which repeat the other half. In the caption, delete words “homologue”. More importantly, are the original concentration data used for Spearman analyses in (b) from this study? Please specify the source of data in the caption. Also, the word “fly ash” is used here and in Supplementary Information-2. Do you consider i-PM and fly ash the same thing?

Fig 5b: The unit is “g/t”, which is not easily understood. Suggest adding explanations in the caption, “grams of i-PM emitted per tonne of production capacity”, if correct.

Fig 6: The unit for the numbers is “g” (gram), which is incorrect. Please clarify it is grams per year, per day, or per hour? If year, please add “annual” in the caption. Also, which year should be specified – 2010 or 2020 or another?

Table 1: All concentrations are reported with precision to 1 pg/g. Some larger numbers have six significant figures! Please re-consider the precision level of the data reported in this paper.

Response to reviewers' comments

Reviewers' comments

Reviewer #1:

General comments:

Polyhalogenated carbazoles (PHCZs) are structural similar to that of polychlorinated dibenzofurans (PCDFs) that attracting global concerns on their sources and control. The toxicity of PHCZs is like PCDFs with high carcinogenicity. However, compared to PCDFs, very little attention was given to the potential sources of PHCZs. The authors conducted many monitoring activities of polyhalogenated carbazoles in particle matters collected from a big size of industrial plants on real full-scale. This study provides essential new data for recognizing important sources of PHCZs and implementing their controlling strategy. I recommend minor revisions before it could be accepted.

Response:

We greatly appreciate your positive comments about the importance of our monitoring data and the constructive suggestions provided. We have carefully revised our manuscript based on your comments. Detailed responses to your comments are provided below.

Specific comments:

Comment (1):

I suggest adding a figure in the supplementary information to display the molecular structure of polyhalogenated carbazoles with numbering of carbon atoms, as well as the structures of the detected congeners in the study. It could help to easily understand their congener profiles.

Response:

Thank you for your comment. We have added the molecular structure of polyhalogenated carbazoles with numbering of carbon atoms (Supplementary Information 1) as well as information about each analyte (Supplementary Information 2), reproduced below.

Supplementary Information 1

Structure of PHCZs with numbering of substitution positions. (X indicates halogen atoms, and m and n indicate their number: $1 \leq m + n \leq 8$).

Supplementary Information 2

Information of PHCZ congeners detected in this research

Analytes	Abbreviation	Cas number	Molecular Formula	Structure
3-Chlorocarbazole	3-CCZ	2372-25-4	$C_{12}H_8ClN$	
3,6-Dichlorocarbazole	3,6-CCZ	5599-71-3	$C_{12}H_7Cl_2N$	
1,3,6,8-Tetrachlorocarbazole	1,3,6,8-CCZ	58910-96-6	$C_{12}H_5Cl_4N$	
2,3,6,7-Tetrachlorocarbazole	2,3,6,7-CCZ	-	$C_{12}H_5Cl_4N$	
3-Bromocarbazole	3-BCZ	1592-95-6	$C_{12}H_8BrN$	
2,7-Dibromocarbazole	2,7-BCZ	136630-39-2	$C_{12}H_7Br_2N$	
3,6-Dibromocarbazole	3,6-BCZ	6825-20-3	$C_{12}H_7Br_2N$	
1,3,6-Tribromocarbazole	1,3,6-BCZ	55119-10-3	$C_{12}H_6Br_3N$	
1,3,6,8-Tetrabromocarbazole	1,3,6,8-BCZ	55119-09-0	$C_{12}H_5Br_4N$	
1-Bromo-3,6-Dichlorocarbazole	1-B-3,6-CCZ	100125-05-1	$C_{12}H_6BrCl_2N$	
1,8-Dibromo-3,6-Dichlorocarbazole	1,8-B-3,6-CCZ	100131-03-1	$C_{12}H_5Br_2Cl_2N$	

Comment (2):

The authors compared the concentrations of PHCZs in different industrial sources and make reasonable explanations for most of the studied industrial sources. However, the low concentrations of PHCZs in lead smelting and coal fired power plants were not explained. The authors should discuss the possible reasons of the low concentrations. In addition, the authors should explain why high concentrations of PHCZs in PM from coke production?

Response:

Thank you for your comment. PHCZs are unintentionally produced persistent organic pollutants (POPs), which might share a similar formation mechanism with that of other POPs, such as dioxins and polychlorinated naphthalenes (PCNs). Secondary lead smelting processes generate much lower concentrations of dioxins and PCNs than other secondary non-ferrous metal smelting processes¹. Previous studies indicated that lead has less catalytic activity than copper and zinc for the formation of PCDD/Fs during smelting processes². Sulfur in coal can inhibit the formation of dioxins during processes of coal fired power plants³. A similar mechanism might be the reason for the low concentrations of PHCZs from coal-fired power plants. Coke production generates coal tar as an important by-product, of which carbazole constitutes 0.5% to 1.8%⁴. The halogenation of carbazole during coke production might lead to the high concentration of PHCZs, which is perhaps why we observe high concentrations of PHCZs in i-PM samples from coke production. The above explanations have been added to the revised manuscript.

L98–101: Carbazole is an important by-product of coke production from coal tar⁴. The possible halogenation of carbazole might lead to highly elevated concentration of PHCZs in i-PM from coke production, making COP a significant but unrecognized industrial source of PHCZs.

L116–121: This is consistent with the previously reported discrepancy in diverse secondary smelting sources of unintentional emissions of polychlorinated naphthalenes, which share similar mechanisms to the unintentional emissions of persistent organic pollutants (POPs)¹. The limited ability of lead to promote the formation of POPs might be the reason for the relatively low Σ PHCZs of i-PM from SPb².

L134–135: Low concentrations of PHCZs have been detected in i-PM samples from 15 CFP facilities owing to the possible inhibitory effect of sulfur in coal³.

Comment (3):

What are the potential release pathways for organic chemical plants? Is it possible for PHCZs to enter environment by waste water, gas diffusion, solid or liquid residues like those of dioxin emissions? Please add relevant discussion.

Response:

Thank you for your comment. The bottom liquid samples collected in our research require further treatment, such as incineration as hazardous wastes. However, we found considerable PHCZs in sediment samples from Yaer Lake, an area suffering from historical wastewater discharge from chemical plants. The results indicated that wastewater is a potential PHCZ release pathway in organic chemical production. The above explanations have been added to the revised manuscript at lines 277–282, with details shown in supplementary information 8.

L277–282: An estimation of PHCZ emission from OC with high concentration in the first two processes investigated (tetrachloroethylene and chlorobenzene production) is not given either because further comprehensive investigations are required. However, our field study of Yaer Lake (Supplementary Information 8) demonstrated the significance of historical chlor-alkali industries (via gas diffusion, solid residues, and wastewater) for environmental PHCZ residues.

Supplementary Information 8 (unpublished data)

Concentrations of PHCZs in sediment from Yaer Lake, an area suffering historical pollution from discharge of organic chemical production

To further determine the PHCZ pollution caused by chemical production, we determined the concentration levels of PHCZs in the sediment of Yaer Lake, historically a key area of chemical production, at different stages.

In 1976, discharge of the chlor-alkali chemical plant led to the accumulation of PAHs, PCDD/Fs, and organochlorine pesticides in lake silt. Dredging work was carried out in 2002 for treatment, and silt was deposited in the soil pile area. By comparing the concentrations of PHCZs in different areas with those in ponds 1–5, a higher concentration was obtained in sediment from the pile soil area, which was formed during the desilting project in 2002. The results indicate that wastewater is an important discharge path of PHCZs in chemical production.

Comment (4):

Are the process techniques adopted in the plants the dominant in each of the investigated industrial categories in China or in the world?

Response:

Thank you for your comment. With reform and opening-up, China has rapidly developed industries in the past several decades. Many advanced techniques in the world have been adopted by Chinese industries, such as coke production, metallurgical plants, and waste incineration. The process techniques in China are currently the mainstream in the world, which is supported by the production scale and advanced technology introduction.

First, the production scale of Chinese industries is ranked high worldwide. For example, China produced 471 million tonnes of coke in 2019, accounting for 69% of the world's coke yield. In 2021, China's electric furnace steel production capacity was approximately 186 million tonnes, representing 33.0% of the world's production. In line with China's plans to promote the circular economy, the output of secondary copper smelting in China was 3.38 million tonnes in 2021, representing 84.5% of the total

copper production in the world (4.00 million tonnes, International Copper Study Group). These results strongly indicate that the process technologies of Chinese industries are mature.

Second, advanced technologies and equipment have been introduced for 40 years since Reform and Opening-up for industry development in China. For example, in 2006, China introduced two 7.63-m coke ovens from Germany's Kaiserstuhl coking plant, opening a new chapter in coking with large-scale coke ovens. With years of development, the world's mainstream coking technology has been successfully adapted in China, such as the coke dust collection system. Moreover, since China's participation in the World Trade Organization, China's municipal waste incineration industry has developed rapidly. By the end of 2020, 463 municipal waste incineration power generation projects in China had been implemented.

Overall, concentrations of PHCZs in i-PM from plants in China are acceptable to represent the world's average level for a first estimation of annual PHCZ emissions through these industries.

Comment (5):

In the Supplementary Information 16, relative effect potencies (REPs) of PHCZs based on the structure-dependent induction of cytochrome P450 1A1 mRNAs in MDA-MB-468 breast cancer cells were not listed. Please list those REPs of PHCZ congeners that adopted for TEQ calculations.

Response:

Thank you for your comment. Riddell et al. determined the aromatic hydrocarbon receptor activation of PHCZs based on CYP1A1 and CYP1B1 gene expression in Ah-responsive MDA-MB-468 breast cancer cells, which was published in ES&T in 2015⁵. The REPs in the research shown below have been added to supplementary information 13.

Supplementary Information 13
The relative effect potencies (REPs) of PHCZs⁵

Congeners	Range of REPs	
	CYP1A1	CYP1B1
2,3,7,8-Tetrachlorodibenzodioxin	1	1
3-Chlorocarbazole	2.7×10⁻⁵	-
3,6-Dichlorocarbazole	1.1×10⁻⁴	-
1,3,6,8-Tetrachlorocarbazole	6.6×10⁻⁴	5.8×10⁻³
2,3,6,7-Tetrachlorocarbazole	1.0×10⁻⁴	3.2×10⁻³
3-Bromocarbazole	1.8×10⁻⁵	2.6×10⁻⁴
2,7-Dibromocarbazole	1.3×10⁻⁵	1.3×10⁻⁴
3,6-Dibromocarbazole	1.7×10⁻⁵	1.5×10⁻⁴
1,3,6-Tribromocarbazole	9.0×10⁻⁵	8.5×10⁻⁴
1,3,6,8-Tetrabromocarbazole	3.1×10⁻⁴	9.7×10⁻³
1-Bromo-3,6-Dichlorocarbazole	6.0×10⁻⁵	3.3×10⁻⁴

Comments (6):

Supplementary Information 3. The information in this section, such as Production Equipment/Scale, Raw Materials should be checked and confirmed. Description about Coke Oven Chamber in the Table seems to be incorrect.

Response:

Thank you for your comment. We checked and confirmed the Industry Coefficient Manual and updated the emission coefficients of particulate matter for these industries, which has been added to supplementary information 9.

Supplementary Information 9**PM emission factors (EFs) of diverse production scales, raw materials, end products, and procedures⁶**

a: CL(T): Coal loading (Top-charging) b: CD(T): Coke Discharging (Top-charging) c: BFGH(T): Blast Furnace Gas Heating (Top-charging) d: DCQ(T): Dry Coke Quenching (Top-charging) e: CL(S): Coal Loading (Stamp-Charging) f: CD(S): Coke Discharging (Stamp-Charging) g: BFGH(S): Blast Furnace Gas Heating (Stamp-Charging) h: WCQ(S): Wet Coke Quenching (Stamp-Charging) i: PMEFs: Emission Factor of PM

Industry	Production Link and Process	Production Scale	Raw Material	Product	PM Producing Coefficient (kg/t)	PM removal equipment	PM removal efficiency	Final PM EFs (g/t)	PMEFs ⁱ for Estimation (g/t)
SCu	Refining	-	Crude Copper	Anode Copper	0.07	Bag Filter	99%	1	479
	Fire Melting	-	Low-grade Copper Scrap	Crude Copper	31.07	Bag Filter	98%	621	
	Pyro-refining	-	How-grade Copper Scrap	Anode Copper	16.72	Bag Filter	98%	334	
	electrolyzation	-	Anode Copper	Cathode Copper	-	Bag Filter	-	-	
	CL(T) ^a				-	Bag Filter	-	102	
	CD(T) ^b				-	Bag Filter	-	129	
	BFGH(T) ^c	Owen Height > 6 m			-	Bag Filter	-	114	
	DCQ(T) ^d				-	Bag Filter	-	105	
	CL(T)	6m >			-	Bag Filter	-	121	
	CD(T)	Owen Height > 4.3 m			-	Bag Filter	-	134	
COP	BFGH(T)				-	Bag Filter	-	119	442
	DCQ(T)		Coking Coal	Coke	-	Bag Filter	-	113	
	CL(T)				-	Bag Filter	-	174	
	CD(T)	4.3m >			-	Bag Filter	-	120	
	BFGH(T)	Owen Height			-	Bag Filter	-	142	
	DCQ(T)				-	Bag Filter	-	124	
	CL(S) ^e				-	Bag Filter	-	115	
	CD(S) ^f				-	Bag Filter	-	131	
	BFGH(S) ^g	-			-	Bag Filter	-	120	
	WCQ(S) ^h				-	Bag Filter	-	68	
EAF		-	Scrap Steel, Pig Iron, Iron Alloy, Slagging Agent	Carbon Steel	26.19	Bag Filter	99.6%	105	105
	Electric Furnace Steelmaking	> 50 t	Scrap Steel, Hot Iron, Ferrochrome, Direct Reduced Iron,	Alloy Steel	13.15	Bag Filter	99.6%	53	
		< 50 t			17.99	Bag Filter	99.6%	72	

		Slagging Agent Scrap Steel, Hot Iron, Ferrochrome, Direct Reduced Iron, Slagging Agent	Stainless Steel					
	-			78.4	Bag Filter	99.6%	314	
	Sintering machines > 360 m ²			5.76	Bag Filter	0.9957	25	
	Sintering Machine Head 180 m ² < sintering machines < 360 m ²			5.81	Bag Filter	0.9957	25	
	sintering machines < 180 m ²			3.45	Bag Filter	0.9957	15	
	sintering machines > 360 m ²			5.96	Bag Filter	0.9957	26	
IOS	Sintering Machine End 180 m ² < sintering machines < 360 m ²	Iron ore, Lime, Coke Powder, Pulverized Coal	Sinter	4.88	Bag Filter	0.9957	21	65
	sintering machines < 180 m ²			3.6	Bag Filter	0.9957	15	
	sintering machines > 360 m ²			6.54	Bag Filter	0.9957	28	
	Discharge 180 m ² < sintering machines < 360 m ²			5.6	Bag Filter	0.9957	24	
	sintering machines < 180 m ²			3.75	Bag Filter	0.9957	16	

Reviewer #2:

General comment:

This is a timely report on the investigation of anthropogenic source of polyhalogenated carbazoles (PHCZs), a new class of emerging contaminants with considerable environmental loads and risk. Although increasing studies have suggested that PHCZs in the environment could be attributable to anthropogenic and natural sources, the formation/emission pathways are still inclusive. This study for the first time investigated the emissions of many previously unrecognized sources of PHCZs from industrial activities. The emissions of PHCZs from 12 large-scale industries were identified and evaluated. The presented source emission data is very abundant and important for interpreting the widespread occurrence of PHCZs in various environments. I recommend to accept this manuscript for publication after minor revisions or explanations.

Response:

Thank you for your positive comments about the timeliness of our work. The suggestions are also very helpful to improve this first investigation of PHCZs emitted from these industrial sources. We further expanded the investigated industrial plants from 26 factories to 122 factories, which allows for a more comprehensive assessment of PHCZ emissions in various industries. Furthermore, we have also determined the emissions of PHCZs from a new source, iron ore sintering, which improves the overall understanding of industrial PHCZ pollution. We have revised the manuscript based on these suggestions. Detailed responses to the specific comments are listed below.

Specific comments:

Comment (1):

Did the authors characterize the fine particle matters collected from the large-scale industrial sources? I suggest providing a few examples (such as 1-3 samples) to display the basic properties of the particle matters from industrial sources, such particle size.

Response:

Thank you for your comment. To determine the basic properties of industrial fine particles, SEM experiments of i-PM samples from all industries were conducted. The particles have diameters of <2.5 μm . SEM results have been added to supplementary information 11.

Supplementary Information 11

Scanning electron microscopy of industrial fine particulate matter samples

Comment (2):

“EAF (electric arc furnace steel)” should be changed to “EAF (electric arc furnace for steel-making)”.

Response:

Thank you for your comment. We have changed “electric arc furnace steel” to “electric arc furnace for steel-making” in the revised manuscript at line 77.

L77: electric arc furnace for steel-making (EAF)

Comment (3):

Eleven PHCZ congeners were detected. There are more congeners than detected in this manuscript. Could the authors further explain why those eleven congeners were selected in this study?

Response:

Thank you for your comment. The PHCZ congeners widely detected in the environment are the 11 PHCZ congeners selected for this study. For example, in the research “*From Sediment to Top Predators: Broad Exposure of Polyhalogenated Carbazoles in San Francisco Bay (U.S.A.)*” published in *ES&T* in 2017, a suite of 11 PHCZ congeners in San Francisco Bay sediment and organisms was investigated⁷. Additionally, in the research “*Occurrence and exposure risk evaluation of polyhalogenated carbazoles (PHCZs) in drinking water*” published in *Science of the Total Environment* in 2021, the same suite of 11 PHCZs in drinking water was reported⁸. Our research is aimed at exploring more potential industrial sources. With this goal, we selected the 11 PHCZs widely detected in the environment to explore the relationship between PHCZs from industrial sources and PHCZs accumulated in the environment and the possible industrial contribution to environmental PHCZ pollution. In our revised manuscript, we discuss why the 11 congeners were selected.

Investigations of 11 PHCZs in the environment

Year	References	Number of Qualified Congeners
2015	Characterization and Biological Potency of Mono- to Tetra- Halogenated Carbazoles	11
2016	Multi-residue determination of polyhalogenated carbazoles in aquatic sediments	11
2017	From Sediment to Top Predators: Broad Exposure of Polyhalogenated Carbazoles in San Francisco Bay (U.S.A.)	11
2017	Polyhalogenated carbazoles in sediments from Lake Tai (China): Distribution, congener composition, and toxic equivalent evaluation	11
2019	Method development for analyzing ultratrace polyhalogenated carbazoles in soil and sediment	11

2019	Analysis of polyhalogenated carbazoles in sediment using liquid chromatography–tandem mass spectrometry	11
2020	Occurrence and distribution of polyhalogenated carbazoles (PHCs) in sediments from the northern South China Sea	11
2020	Determination of polyhalogenated carbazoles in soil using gas chromatography–triple quadrupole tandem mass spectrometry	11
2021	Polyhalogenated carbazoles in freshwater and estuarine sediment from China and the United States: A multi-regional study	11
2021	Multiple classes of chemical contaminants in soil from an e-waste disposal site in China: Occurrence and spatial distribution	11
2021	Occurrence and exposure risk evaluation of polyhalogenated carbazoles (PHCZs) in drinking water	11

L80–84: Industrial fine particulate matter (i-PM) and chemical bottom liquids were analyzed to identify the 11 PHCZ congeners frequently detected in the environment (Supplementary Information 2). PHCZ congener profiles of industrial samples were compared with those of environmental samples (Supplementary Information 3).

Supplementary Information 3

Environmental occurrences of PHCZs cited in this research^{7,9–18}

Environmental occurrences and references	Location
Sedi-1 ⁹	Northern Germany
Sedi-2 ⁷	San Francisco Bay, USA
Sedi-3 ¹¹	Lake Tai, China
Sedi-4 ¹²	Zhou Shan, China
Sedi-5 ¹³	Qingdao, China
Soil-1 ¹²	Taizhou, China

Soil-2 ¹⁴	Zhoucheng, China
Soil-3 ¹⁵	Northeast Provinces, China
Soil-4 ¹⁵	Northeast Provinces, China
Soil-5 ¹⁵	Northeast Provinces, China
Soil-6 ¹⁵	Northeast Provinces, China
Soil-7 ¹⁵	Northeast Provinces, China
Soil-8 ¹⁶	Tibetan Plateau, China
Soil-9 ¹⁷	Hangzhou, China
Dust-1 ¹⁸	Derenda, Teltow, Germany
Dust-2 ¹⁰	Hangzhou, China

Comment (4):

The authors calculated the toxic equivalents (TEQs) of PHCZ emissions. What are the toxic equivalent factors (TEFs) used for calculating the TEQs of PHCZs? Are those TEFs widely used? Are they similar to those of dioxin congeners with TEFs issued by WHO?

Response:

Thank you for your comment. To the best of our knowledge, there are no available TEFs issued by official organizations, such as WHO. The relative effect potencies (REPs) of PHCZs used in our calculations of TEQs are based on the REPs from Riddle’s research, “*Characterization and Biological Potency of Mono- to Tetra- Halogenated Carbazoles*”, in which the aromatic hydrocarbon receptor activation of PHCZs was determined on the basis of CYP1A1 and CYP1B1 gene expression in Ah-responsive MDA-MB-468 breast cancer cells⁵. The set of REPs of PHCZs is widely accepted and used in later research, such as “*Polyhalogenated carbazoles in sediments from Lake Tai (China): Distribution, congener composition, and toxic equivalent evaluation*” published in *Environmental Pollution*¹¹, “*From Sediment to Top Predators: Broad Exposure of Polyhalogenated Carbazoles in San Francisco Bay (U.S.A.)*”⁷ and “*Bioaccumulation and Spatiotemporal Trends of Polyhalogenated Carbazoles in Great Lakes Fish from 2004 to 2016*”¹⁹ published in *ES&T*, and “*Occurrence and exposure risk evaluation of polyhalogenated carbazoles (PHCZs) in drinking water*” published in *Science of the Total Environment*⁸. The REFs of PHCZs are shown below and also in supplementary information 13.

Supplementary Information 13

The relative effect potencies (REPs) of PHCZs⁵

Congeners	Range of REPs	
	CYP1A1	CYP1B1
2,3,7,8-Tetrachlorodibenzodioxin	1	1

3-Chlorocarbazole	2.7×10^{-5}	-
3,6-Dichlorocarbazole	1.1×10^{-4}	-
1,3,6,8-Tetrachlorocarbazole	6.6×10^{-4}	5.8×10^{-3}
2,3,6,7-Tetrachlorocarbazole	1.0×10^{-4}	3.2×10^{-3}
3-Bromocarbazole	1.8×10^{-5}	2.6×10^{-4}
2,7-Dibromocarbazole	1.3×10^{-5}	1.3×10^{-4}
3,6-Dibromocarbazole	1.7×10^{-5}	1.5×10^{-4}
1,3,6-Tribromocarbazole	9.0×10^{-5}	8.5×10^{-4}
1,3,6,8-Tetrabromocarbazole	3.1×10^{-4}	9.7×10^{-3}
1-Bromo-3,6-Dichlorocarbazole	6.0×10^{-5}	3.3×10^{-4}
1,8-Dibromo-3,6-Dichlorocarbazole	3.2×10^{-4}	9.7×10^{-3}

Comment (5):

I suggest the author providing a figure about the congener profiles of PHCZs formed during production of indigo dyes and photoelectric materials in the supplementary information. It could provide a visual comparison of different congener profiles from different industrial activities.

Response:

Thank you for your comment. Actually, there are no reports on specific congener profiles of PHCZs generated during the production of indigo dyes and photoelectric materials, although there are reports on specific PHCZ congeners²⁰. The profiles of PHCZs detected in indigo dyes are shown below. We have added descriptions to the **Introduction** section.

Redacted

(From reference: Parette, R. et al., Chemosphere, 2016, 150, 414–415.)

L60–62: However, intermediates of the production of photoelectric materials are 2,7-BCZ and 3,6-BCZ, which have relatively low concentrations in environmental media.

L64–65: Only 1,3,6,8-CCZ, 1,3,6,8-BCZ, and 1,8-B-3,6-CCZ were main impurities in historical synthetic dye²⁰.

Comment (6):

High concentrations of PHCZs in bottom residue liquid samples from organic chemical production were detected. However, the description about the information of production amounts of investigated chemicals on national scale or global scale is inadequate. I suggest supplementing basic information about the production amounts of investigated chemicals for better judging the potential importance of chemical production sources.

Response:

Thank you for your comment. According to data released by the International Chemical Industry Association, the global production of chemicals in 2020 reached 529 million tonnes, the global production of vinyl chloride in 2019 was 48 million tonnes, and the global production of chlorobenzene in 2016 was approximately 3 million tonnes. We have added relevant discussions to our revised manuscript²¹.

L321–323: There is also possible extra emissions through OC, which produces 529 million tonnes of chemicals in 2020²¹.

Comment (7):

Could the authors provide a brief description about the temporal trend of PHCZ emissions from the important industrial sources? For example, recent five or ten years.

Response:

Thank you for your comment. In our research, the global PHCZ emission depended on the production output of these industries. However, the production output of important industries remains stable in the past few years. For example, global production outputs of electric arc furnace for steel-making were 563, 492, 523, 524, and 409 million tonnes in 2021, 2020, 2019, 2018, and 2017, respectively. Moreover, global production outputs of secondary copper production were 4.0, 4.1, 3.8, and 4.0 million tonnes from 2021 to 2018. Although there are no available production output values for global coke production, China, the biggest producer of coke, has produced a stable yield in the past few years.

Figure Global production output of second copper smelting from 2017 to 2021

Figure Global production output of electric arc furnace steelmaking from 2017 to 2021

Figure Production output of coke production in China from 2018 to 2022

Therefore, we considered that the temporal trend of PHCZ emissions is similar to the temporal trend of production output in each industry. However, we agree with the reviewer in that evaluating the historical emission and accumulation of PHCZs over a long timeframe is necessary. We have highlighted this issue in the conclusion to call for more attention.

Line 352–353: (4) field studies or model-based to quantify PHCZ emissions from historical industrial activities;

Comment (8):

Line 65: Cl-/Br- instead of Cl/Br.

Response:

Thank you for your comment. We have changed this in the revised manuscript. The modified content is as follows:

Line 52–53: Fungal activity can be a natural PHCZ source. Mumbo et al.²² confirmed that in the presence of H₂O₂ and Cl-/Br-, fungi can bio-transform carbazole (CZ) to PHCZs.

Comment (9):

Line 107: Change “detection rate” to “detection frequency” and throughout.

Response:

Thank you for your comment. We have made this change in the revised manuscript. The modified content is as follows:

Line 90–91: Material samples potentially with PHCZs originated in the following industries: OC, CK, MSWI, HWI, COP, EAF, IOS, PCu, SAl, SCu and SZn.

Comment (10):

Line 148-149: The formation of PCDFs from the coupling reaction of chlorophenol and chlorobenzene under high temperature has been well studied. However, it is unclear whether PHCZs can be produced via similar pathways. Could the authors expand more on the possible formation pathways of PHCZs from these industrial activities?

Response:

Thank you for your comment. The *Toolkit for Identification and Quantification of Releases of Dioxins, Furans and Other Unintentional POPs* issued by UNEP and the Stockholm Convention²³ (<http://www.pops.int/Implementation/UnintentionalPOPs/ToolkitforUPOPs/Overview/tabid/372/Default.aspx>) indicate that chemical production is a significant source of PCDFs. PHCZs are similar to PCDFs in structure and was detected in high concentrations in the same source as PCDFs. The coupling reaction of chlorophenol and chlorobenzene is considered a formation pathway of PCDFs. However, some studies suggest that nitrobiphenyl is an important precursor for synthesizing bromocarbazole, and the method for synthesizing 2-bromocarbazole with 2-nitro-4-bromobiphenyl as a precursor is mature. In 2014, ELMABRUK²⁴ used 2-nitro-4-bromobiphenyl as a raw material, which was refluxed at 170 ° C for 12 h to obtain 89% of 2-bromocarbazole. Some patents have documented the synthesis of bromocarbazole through 2-nitro-4-bromobiphenyl.

Synthesis path of 2-bromocarbazole

A similar formation pathway of PHCZs might exist in chemical production. Screening of the bottom liquid sample (OC-1) with high resolution Q-TOF mass spectrometry showed that two N-containing compounds, nitrobenzene and $C_9H_{14}ClN_3O$, existed in the bottom liquid. A benzene series, such as biphenyl and chlorobenzene, was also identified. These compounds might generate chlorinated nitrobiphenyls, leading to the production of PHCZs.

We have included relevant content in the revised manuscript and screening results in supplementary information 5.

L136–143: High concentrations of PHCZs have been detected in matrices collected from chemical manufacturing plants. Previous studies have demonstrated that the nitrobiphenyl group is an important precursor of bromocarbazole synthesis^{24,25}. The existence of nitrobenzene, biphenyl, and chlorobenzene in material from chemical manufacturing plants was confirmed using high resolution Q-TOF mass spectrometry screening (Supplementary Information 5). The co-existence of multiple precursors might lead to the formation of significant amounts of PHCZs during chemical manufacturing processes.

Supplementary Information 5

No-target screening of OC-1 using Q-TOF mass spectrometry

Name	CAS	Match score	Formula
Benzene, 1,2-dichloro-	95-50-1	90.2	$C_6H_4Cl_2$
Benzene, 1,3-dichloro-	541-73-1	94	$C_6H_4Cl_2$
Cyclohexene, .gamma.-3,4,5,6-tetrachloro-	319-81-3	79.5	$C_6H_6Cl_4$
Benzene, 1,4-dichloro-	106-46-7	95	$C_6H_4Cl_2$
1,1'-Biphenyl, 4-chloro-	2051-62-9	98.5	$C_{12}H_9Cl$
1,1'-Biphenyl, 4-chloro-	2051-62-9	98.7	$C_{12}H_9Cl$
1,1'-Biphenyl, 3,3'-dichloro-	2050-67-1	97.1	$C_{12}H_8Cl_2$
1,1'-Biphenyl, 3,3'-dichloro-	2050-67-1	97.5	$C_{12}H_8Cl_2$
Benzene, 1,2,3-trichloro-	87-61-6	98.8	$C_6H_3Cl_3$
m-Terphenyl	92-06-8	98.7	$C_{18}H_{14}$

1,1'-Biphenyl, 2,2'-dichloro-	13029-08-8	97.1	C ₁₂ H ₈ Cl ₂
Benzene, 1-chloro-4-methyl-	106-43-4	97.9	C ₇ H ₇ Cl
.delta.-Pentachlorocyclohexene	643-15-2	93	C ₆ H ₅ Cl ₅
Phenol, 2-chloro-	95-57-8	97.8	C ₆ H ₅ ClO
.delta.-Pentachlorocyclohexene	643-15-2	94.3	C ₆ H ₅ Cl ₅
1,1'-Biphenyl, 3,3'-dichloro-	2050-67-1	97.8	C ₁₂ H ₈ Cl ₂
.alpha.-Lindane	319-84-6	96.7	C ₆ H ₆ Cl ₆
Biphenyl	92-52-4	98.2	C ₁₂ H ₁₀
Benzene, 1-bromo-4-chloro-	106-39-8	98.7	C ₆ H ₄ BrCl
m-Terphenyl	92-06-8	97.9	C ₁₈ H ₁₄
1,1'-Biphenyl, 2-methyl-	643-58-3	97.4	C ₁₃ H ₁₂
1,1'-Biphenyl, 3,3'-dichloro-	2050-67-1	96.8	C ₁₂ H ₈ Cl ₂
Cyclopentane, (trichloroethenyl)-	55255-41-9	75	C ₇ H ₉ Cl ₃
1,1'-Biphenyl, 4-chloro-	2051-62-9	96.7	C ₁₂ H ₉ Cl
p-Terphenyl, 2,5-dichloro-	61576-83-8	92.5	C ₁₈ H ₁₂ Cl ₂
1,1'-Biphenyl, 4-methyl-	644-08-6	97.4	C ₁₃ H ₁₂
5,5,10,10-Tetrachlorotricyclo[7.1.0.0(4,6)]decane	17725-81-4	61.9	C ₁₀ H ₁₂ Cl ₄
1-Heptene, 5,7,7,7-tetrachloro-	51287-99-1	57.3	C ₇ H ₁₀ Cl ₄
p-Terphenyl, 4-chloro-	1762-83-0	77.1	C ₁₈ H ₁₃ Cl
.alpha.-Lindane	319-84-6	92	C ₆ H ₆ Cl ₆
1,1'-Biphenyl, 3,3'-dichloro-	2050-67-1	95.9	C ₁₂ H ₈ Cl ₂
3-Chlorodiphenylmethane	27798-38-5	95.5	C ₁₃ H ₁₁ Cl
Benzene, nitro-	98-95-3	92.5	C ₆ H ₅ NO ₂
Pyridazin-3(2H)-one, 5-chloro-2-methyl-4-(2-methylpropylamino)-	98795-97-2	70.8	C ₉ H ₁₄ ClN ₃ O

Comment (11):

As shown in Figure 1, high concentrations of PHCZs are emitted from coke production, secondary aluminum smelting, and municipal solid waste incineration. However, the emitting PHCZ concentrations from the three activities varied greatly from different plants. As for the same industry, are these plants used the same raw material and process for production? Could the authors explain why? What are the concentrations used for the estimation of the global emissions of PHCZs from these industries, since the emitting concentration of PHCZs varied greatly with different plants.

Response:

Thank you for your comment. The differences in raw materials might be the reason for the variability. A previous study reported that raw materials play a significant role in the amount of generated PCDD/Fs. For example, the raw material of secondary aluminum smelting, aluminum scrap, can be scrapped in different application scenarios, leading to differences in halogen and carbon contents. Therefore, there might be uncertainty in industrial field studies.

However, we tried to reduce the uncertainty for our revised manuscript. We made efforts to expand the number of investigated plants from 26 to 122 factories, almost a 5-fold increase. At least six plants

were included in each industry for a more comprehensive assessment of PHCZs emission. We excluded minimaxes through box plots. The mean of the remaining values (5–13 figures) in each industry was used to estimate the emissions.

Because of the difficulties in sample acquisition from industrial plants and high cost of laboratory analysis, investigations of large numbers of industrial samples are lacking. Previous studies^{26,27} used FT-ICR mass spectrometry with only one or two samples for molecular characterization of organic chemicals in important industries. In another study, three gas samples collected from the boiler outlet, BF outlet, and stack in one plant were applied to investigate dioxin-like PCBs released from municipal solid waste incineration into the atmosphere²⁸. In the *Toolkit for Identification and Quantification of Releases of Dioxins, Furans and Other Unintentional POPs* issued by UNEP and the Stockholm Convention, the emission factor is the average of the emission factors from only several plants. Relevant revisions are shown below and have been added to the revised manuscript.

L74: Here, we conducted field investigations of 13 industries, covering 122 factories.

L359–361: In total, 122 industrial plants from 13 industries were included in this study. This comprehensive field investigation enables source identification and emission estimation of POPs in one single study²³.

L448–451: More than six industrial plants were sampled for each industry (Supplementary Information 4). To reduce the variation associated with diverse plants, the box plot was used to exclude the minimax of Σ PHCZs in each industry, and then the average Σ PHCZs of diverse factories was used.

Supplementary Information 7

Data processing method for characteristics of PHCZ concentrations, congener profile analysis, principal component analysis, and emission estimation

Characteristics of PHCZ concentrations: For industrial sources, the average concentration of each congener in i-PM samples from all factories in a specific industry is used for the calculation of Spearman's coefficients among 11 congeners in i-PM samples. For environmental occurrences, the average concentration of each congener in PHCZ investigation of soil or sediment is used for the calculation of Spearman's coefficients among 11 congeners in environmental media. Data analysis is conducted using SPSS.

Congener profile analysis: For industrial sources, the average proportion of each congener in i-PM samples from all factories in the specific industry is used to represent the PHCZ congener profile of this industry. For environmental occurrences, the average proportion of each congener in PHCZ investigation of soil or sediment is used to represent the PHCZ congener profile in this environmental matrix.

Principal component analysis: principal component analysis is conducted for 120 i-PM samples and 16 environmental occurrences based on individual congener profile of PHCZs. Data analysis is conducted using SPSS.

Emission estimation: To reduce the variance among total PHCZ concentrations of i-PM samples in each industry, minimaxes are removed through the box plot, which is located at $\square 1.5$ inter-quartile range (IQR). The remaining PHCZ concentrations are averaged to obtain the PHCZ content of i-PM samples from each industry.

Comment (12):

What is data source of the PHCZs in the environment presented in Figure 2? My understanding is that the figure analyzing the correlation of PHCZ congeners between industrial particle matters and environmental samples. How can these results be used to interpret the occurrence of PHCZs in the environment?

Response:

Thank you for your comment. In this research, the data are from previous environmental investigations of PHCZs. We have added descriptions and details to supplementary information 3. We checked the figures and made some correction. Some figures have been added to describe the correlations of PHCZ congener concentrations in each industry and to present the emission features of each industry. The same analysis was also conducted for PHCZs in soil and sediment, two environmental matrices containing PHCZs, which combined with congener concentrations was used for source identification of specific congeners. We have provided an interpretation of these results in the revised manuscript.

L82–84: PHCZ congener profiles of industrial samples were compared with those of environmental samples (Supplementary Information 3).

L190–218: Spearman analysis of the congener concentrations and environmental occurrences of PHCZs produced by the investigated industries (see Supplementary Information 6 and Supplementary Information 3) was conducted to understand the source and environmental

characteristics of PHCZs (data processing in Supplementary Information 7). As shown in Fig. 3 (red and blue indicate positive and negative correlations, respectively), certain congeners display a strong positive correlation with other congeners in SAI, MSWI, and CK. However, weak correlations with other congeners could be seen for 3-CCZ and 3-BCZ in SAI, 1,3,6-BCZ, 1,3,6,8-BCZ, and 2,3,6,7-CCZ in MSWI, and 2,3,6,7-CCZ in CK. In the other industries, correlations exist between minority of congeners. For example, in COP, SZn, and IOS, positive correlations exist among 3-CCZ, 3-BCZ, 3,6-CCZ, and 1,3,6,8-CCZ. In SCu, positive correlations exist among 2,7-BC, 3,6-BCZ, 1,8-B-3,6-CCZ, and 1,3,6-BCZ, which is a distinguishing feature among secondary metal smelting industries. Interestingly, strong negative correlations between certain congeners in the environment are absent in every industry. For example, 3,6-CCZ has negative correlations with 10 other congeners in sediment and soil but weak or positive correlations with these congeners in all industries. The proportion of 3,6-CCZ in i-PM samples is high but the correlations between 3,6-CCZ and other congeners in the environment and industries differ, suggesting that anthropogenic and natural sources might jointly contribute to 3,6-CCZ pollution. The concentrations of 1-B-3,6-CCZ, 1,8-B-3,6-CCZ, 1,3,6-BCZ, 1,3,6,8-BCZ, and 2,3,6,7-CCZ are low, while the concentrations of 2,7-BCZ and 3,6-BCZ are high in some i-PM samples, such as from COP-1 and SAI-1 (Supplementary Information 6). 3-CCZ, 3-BCZ, 3,6-CCZ, and 1,3,6,8-CCZ exist in high concentration in i-PM samples from most industries, indicating that industrial activities are part of their sources. The results from environmental investigation of PHCZs support our findings. For example, Guo et al.²⁹ found that in sediments from Great Lakes, accumulated 1,3,6,8-BCZ is from natural sources, 3,6-CCZ residue arises from both anthropogenic and natural sources, and accumulated 1,3,6,8-CCZ in fish is from anthropogenic sources¹⁹.

Supplementary Information 3

Environmental occurrences of PHCZs cited in this research^{7,9-18}

Environmental occurrences and references	Location
Sedi-1 ⁹	Northern Germany
Sedi-2 ⁷	San Francisco Bay, USA
Sedi-3 ¹¹	Lake Tai, China
Sedi-4 ¹²	Zhou Shan, China
Sedi-5 ¹³	Qingdao, China
Soil-1 ¹²	Taizhou, China
Soil-2 ¹⁴	Zhoucheng, China
Soil-3 ¹⁵	Northeast Provinces, China
Soil-4 ¹⁵	Northeast Provinces, China
Soil-5 ¹⁵	Northeast Provinces, China
Soil-6 ¹⁵	Northeast Provinces, China
Soil-7 ¹⁵	Northeast Provinces, China
Soil-8 ¹⁶	Tibetan Plateau, China
Soil-9 ¹⁷	Hangzhou, China
Dust-1 ¹⁸	Derenda, Teltow, Germany
Dust-2 ¹⁰	Hangzhou, China

Reviewer #3 (Remarks to the Author):

General comment:

This paper, for the first time, reports the concentrations of PHCZs in particulate matter (and bottom liquid) generated from industries. A total of 26 samples were collected from 26 plants of eight categories of industry, and 11 PHCZ congeners were analyzed in each sample. Efforts were made to examine the congener pattern, explore correlations between industrial emissions and environmental occurrence, and estimate the global industrial release of PHCZs. Such work is clearly important because, as the authors mentioned in L82-84 and I agree, that “current research on source identification is woefully inadequate, and knowledge to date does not support any reasonable explanation of the huge residues, wide distribution and complicated congener profiles of PHCZs in the environment”. To this end, the necessity of conducting this work is justified. I also consider the experimental dataset valuable and would like to encourage its publication in an environmental journal. However, for the ambitious goal of providing a global assessment, the amount of work is too small, the conclusions are not solid, and the overall findings are too weak. For this reason, I do not recommend the publication of work in Nature Communications.

Response:

Thank you for your comment. We sincerely appreciate the reviewer’s positive comment about the first-time report of new industrial sources of PHCZs and their emission estimation. We also agree with the reviewer that this study can be improved, specifically the limited number of investigated industrial plants for the original manuscript. To reduce the uncertainty and strengthen the findings and conclusions, we have made great efforts to extend the number of investigated industrial plants.

For the revised manuscript, the number of investigated industrial plants was extended from the original 26 plants to 122 industrial plants. At least six plants were included for each industry for a more comprehensive assessment of PHCZ emissions. To provide more cautious estimations, instead of using the average concentration of all PHCZ congeners in industrial particulate particles to estimate emissions directly, we have excluded minimaxes through box plots. The mean of the remaining values (5-13 figures) in each industry was used to estimate the emissions.

Unlike sampling of environmental samples, there are great difficulties in sample acquisition from industrial plants. It is feasible to collect environmental samples (such as soil or water) according to the spatial distribution of the experimental design with relatively free experimental implementation and relying only on the research group. However, as for industrial sampling, there are great difficulties in sample acquisition from industrial plants. It is essential to communicate, negotiate, and obtain permission for sample collection from the investigated industrial plants. In the *Toolkit for Identification and Quantification of Releases of Dioxins, Furans and Other Unintentional POPs* issued by UNEP and the Stockholm Convention, the industrial emission factor of PCDD/Fs is the average of the emission factors

from only several plants. To our best knowledge, the current sample size with 122 investigated industrial plants in one single field study is very large, especially for PHCZs, which lack industrial field data. We sincerely hope the reviewer understands the practical difficulties when conducting industrial field studies. We also changed the title to '*Polyhalogenated carbazoles: Identification of new sources and estimation of global emissions*' in the revised manuscript. Detailed responses to specific comments are shown below. Additions and revisions are highlighted in the revised manuscript.

Specific comments:

Comment (1):

Natural production of PHCZs is not the focus of this paper, but its importance must not be downplayed. In this paper, natural generations of PHCZs from fungi and volcano are mentioned in two sentences (L64-68), ignoring potentially significant (some are currently unknown) others. Based on this work, the total global emission of PHCZs from the eight industries is only about 10 kg/yr. How does this emission estimate compare with that for dioxins from the same industries? Such a low emission, if correct, could mean that industrial emissions be insignificant compared with natural production, as far as total PHCZs concerned.

Response:

Thank you for your comment. As the reviewer mentioned, this study is focused on the industrial emissions of PHCZs, not the natural production of PHCZs. It was believed that the discovery and estimation of PHCZs from several important industrial activities, first conducted in the study, could greatly contribute to filling the recognition gap of emission sources of PHCZs.

We also fully agree with the reviewer in that natural production contributes to the occurrences of PHCZs in the environment, which has been previously reported, and we have pointed this out in the "Conclusion and perspective" section of the revised manuscript. However, recognition of the sources of PHCZs is still lacking, which needs to be addressed with comprehensive studies of the sources. Fungal activity is a known natural source of PHCZs, and other natural sources are seldom reported. Exploration of other natural sources are also urgently needed to explain PHCZ pollution. Therefore, we have added text to highlight the importance of natural sources. Additionally, we have pointed out that investigations of natural sources of PHCZs should be a research focus in our "Conclusion and Perspective" section to call for more investigations into natural sources of PHCZs.

The significance of natural sources of PHCZs has been discussed and emphasized in the revised manuscript. We have also added relevant descriptions about the importance of natural sources of PHCZs in the “Conclusion and perspective” section of the revised manuscript (as shown below).

According to data from the US EPA (<https://www.epa.gov/trinationalanalysis/dioxins>), global emissions of dioxins in 2020 amounted to 99.07 kg, with 60% originating in emissions from chemical production, 26% from hazardous waste management, and 13% from primary metal smelting. In comparison, the results of this study showed that the 6 important industries emit 13 kg of PHCZs in total, indicating that these industries are capable of emitting a significant amount of PHCZs compared with that of dioxins. Additionally, chemical production was demonstrated to be an important source of PHCZs in the environment (supplementary information 8). Therefore, emissions from industrial activities cannot be ignored. Generally, considering that the toxicity of PHCZs is similar to that of dioxins, which are regulated by the Stockholm Convention, the emissions, regulation, and control of PHCZs from on-going industrial activities in the world also deserve our attention.

Addition and revision in the revised manuscript:

L52–58: Fungal activity can be a natural PHCZ source. Mumbo et al.²² confirmed that in the presence of H₂O₂ and Cl-/Br-, fungi can bio-transform carbazole (CZ) to PHCZs. The load of 1,3,6,8-BCZ and other 12 unknown PHCZs constituting 64% of more than 3000 tonnes of PHCZ residues in sediments from Great Lakes is concluded to be generated from natural activity, implying on the significant natural formation of PHCZs²⁹. It is widely believed that anthropogenic activities also play a role in PHCZ pollution of the environment^{7,11}.

L206-209: The proportion of 3,6-CCZ in i-PM samples is high but the correlations between 3,6-CCZ and other congeners in the environment and industries differ, suggesting that anthropogenic and natural sources might jointly contribute to 3,6-CCZ pollution.

L214-218: The results from environmental investigation of PHCZs support our findings. For example, Guo et al.²⁹ found that in sediments from Great Lakes, accumulated 1,3,6,8-BCZ is from natural sources, 3,6-CCZ residue arises from both anthropogenic and natural sources, and accumulated 1,3,6,8-CCZ in fish is from anthropogenic sources¹⁹.

L353-355: (5) comprehensive investigations of natural PHCZ sources to clarify their significance for PHCZ pollution.

Comment (2):

Where were all the samples collected? Is any from outside China? How are the samples (each from one filter bag used for 48 hr) representative even within China? Additionally, each industry was represented by only 1-4 samples. L461-462 mentions that 2 to 4 samples from each industry are “comparable to the open guidance published by some authorities”, but these open guidelines are not provided. Within each industry, the reported concentrations differ by orders of magnitude. For example, global emissions of 8650 g PHCZs from coke production was estimated from only two COP samples which differ by 11 times in concentration (the first two rows in Table 1). Similarly, the only three samples of MSWI ranged from 0.1 to 29 ng/g (Table 1). The averages were used in the estimations for global emissions in this paper. Such estimates bear huge uncertainties which diminish the differences among industries and make the conclusions in this paper (for example, “The biggest emitter among the investigated industries was COP” in L289) very doubtful.

Response:

Thank you for your comment. In our research, because of the difficulty in cross-border sample collection, all samples were collected from plants in China. However, considering the commonality of global industrial production technologies and China’s major production status in these industries (**Information 1**), the results of PHCZs in samples collected from 122 industrial plants in China are considered significant for filling the recognition gap on unintentional PHCZ emissions from industrial sources.

Regarding the limited number of investigated plants, we have made great efforts to expand the number of investigated industrial plants from the original 26 to 122, covering 13 categories with on-going and massive industrial activities in the world. At least six plants were included for each industry and over 10 plants were investigated for high-polluting industries, such as coke production. Minimaxes of total PHCZ concentrations of i-PM samples in each industry were removed through the box plot, which was located at ± 1.5 inter-quartile range (IQR). The remaining PHCZ concentrations were averaged to obtain the PHCZ content of i-PM samples from each industry for global emission estimation. Usually, industrial field studies suffer from difficulties in sample acquisition (**Information 2**). To our best knowledge, the increased data scale is larger than that of other industrial field studies, especially industrial field studies of PHCZs (**Information 2**). Moreover, in the *Toolkit for Identification and Quantification of Releases of Dioxins, Furans and Other Unintentional POPs* issued by UNEP and the Stockholm Convention (<http://www.pops.int/Implementation/UnintentionalPOPs/ToolkitforUPOPs/Overview/tabid/372/Default.aspx>), the emission factor is derived from only several plants.

Information 1

First, the production scale of Chinese industries is ranked high worldwide. For example, China produced 471 million tonnes of coke in 2019, accounting for 69% of the world's coke yield. In 2021, China's electric furnace steel production capacity was approximately 186 million tonnes, representing 33.0% of the world's production. In line with China's plans to promote the circular economy, the output of secondary copper smelting in China was 3.38 million tonnes in 2021, representing 84.5% of the total production in the world (4.00 million tonnes, International Copper Study Group). These results strongly indicate that the process technologies of Chinese industries are mature.

Second, advanced technologies and equipment have been introduced for 40 years since Reform and Opening-up for industry development in China. For example, in 2006, China introduced two 7.63-m coke ovens from Germany's Kaiserstuhl coking plant, opening a new chapter in coking with large-scale coke ovens. With years of development, the world's mainstream coking technology has been successfully adapted in China, such as the coke dust collection system. Moreover, since China's participation in the World Trade Organization, China's municipal waste incineration industry has developed rapidly. By the end of 2020, 463 municipal waste incineration power generation projects in China had been implemented.

Information 2

As an industrial field study, online monitoring of PHCZs has not been achieved, although online monitoring of normal pollutants (such as NO_x and SO_x) has been conducted. Analysis of PHCZs requires a series of complex analytical procedures. The cost required to identify and quantify PHCZ congeners is also relatively high owing to use of ¹³C-labeled standards. Additionally, unlike sampling of environmental samples, which is relatively easy because it is feasible to collect samples with relatively free experimental implementation and relying only on the research group, there are great difficulties in sample acquisition from industrial plants. It is essential to communicate, negotiate, and obtain permission for sample collection from the investigated industrial plants. All of these factors determine the sample size, which is generally small in industrial POP studies. Previous studies^{26,27} used using FT-ICR mass spectrometry with only one or two samples for molecular characterization of organic chemicals in important industries. In another study, three gas samples collected from the boiler outlet, BF outlet, and stack in one plant were applied to investigate dioxin-like PCBs released from municipal solid waste incineration into the atmosphere²⁸. In the *Toolkit for Identification and Quantification of Releases of Dioxins, Furans and*

Other Unintentional POPs issued by UNEP and the Stockholm Convention²³, the emission factor is the average of emission factors from only several plants.

To our best knowledge, the current sample size with 122 investigated industrial plants in one single field study is large, especially for PHCZs, which lack industrial field data. Therefore, we sincerely hope the reviewer understands the practical difficulties when conducting industrial field studies.

L359–361: In total, 122 industrial plants from 13 industries were included in this study. This comprehensive field investigation enables source identification and emission estimation of POPs in one single study²³.

Comment (3):

The conclusion that “the congeners in the industrial emissions and environmental occurrences exhibit similar grouping” (L337-338) need more caution. The idea that positive correlations indicate cause-effect relationship is wrong. The section starting in L200 is simply a correlation analysis among congeners, it does not support the concluding statement “industrial activities are sources of PHCZs” (L216) “in the world” (L200). Similar comment can be made on the section starting in L218. Congener 1368-BCZ is believed to come from legacy dye industry, which was not involved in this work; thus L238-246 is not needed.

Response:

Thank you for your comment. We have deleted the sentence “the congeners in the industrial emissions and environmental occurrences exhibit similar grouping” and have given a conclusion with more caution in L190-218. We have checked the correlation analysis and updated our conclusions, which focus more on the features of PHCZs through these investigated industries. The sentences that the reviewer pointed out have been rewritten in a more cautious way.

L190–218: Spearman analysis of the congener concentrations and environmental occurrences of PHCZs produced by the investigated industries (see Supplementary Information 6 and Supplementary Information 3) was conducted to understand the source and environmental characteristics of PHCZs (data processing in Supplementary Information 7). As shown in Fig. 3 (red and blue indicate positive and negative correlations, respectively), certain congeners display a strong positive correlation with other congeners in SAI, MSWI, and CK. However, weak correlations with other congeners could be seen for 3-CCZ and 3-BCZ in SAI, 1,3,6-BCZ, 1,3,6,8-BCZ, and 2,3,6,7-CCZ in MSWI, and 2,3,6,7-CCZ in CK. In the other industries, correlations exist

between minority of congeners. For example, in COP, SZn, and IOS, positive correlations exist among 3-CCZ, 3-BCZ, 3,6-CCZ, and 1,3,6,8-CCZ. In SCu, positive correlations exist among 2,7-BC, 3,6-BCZ, 1,8-B-3,6-CCZ, and 1,3,6-BCZ, which is a distinguishing feature among secondary metal smelting industries. Interestingly, strong negative correlations between certain congeners in the environment are absent in every industry. For example, 3,6-CCZ has negative correlations with 10 other congeners in sediment and soil but weak or positive correlations with these congeners in all industries. The proportion of 3,6-CCZ in i-PM samples is high but the correlations between 3,6-CCZ and other congeners in the environment and industries differ, suggesting that anthropogenic and natural sources might jointly contribute to 3,6-CCZ pollution. The concentrations of 1-B-3,6-CCZ, 1,8-B-3,6-CCZ, 1,3,6-BCZ, 1,3,6,8-BCZ, and 2,3,6,7-CCZ are low, while the concentrations of 2,7-BCZ and 3,6-BCZ are high in some i-PM samples, such as from COP-1 and SAI-1 (Supplementary Information 6). 3-CCZ, 3-BCZ, 3,6-CCZ, and 1,3,6,8-CCZ exist in high concentration in i-PM samples from most industries, indicating that industrial activities are part of their sources. The results from environmental investigation of PHCZs support our findings. For example, Guo et al.²⁹ found that in sediments from Great Lakes, accumulated 1,3,6,8-BCZ is from natural sources, 3,6-CCZ residue arises from both anthropogenic and natural sources, and accumulated 1,3,6,8-CCZ in fish is from anthropogenic sources¹⁹.

L225–226: Comparison of congener and homologue patterns between i-PM and environmental samples

L248–250 The preponderance of 3-CCZ, 3-BCZ, and 3,6-CCZ in both i-PM and environmental samples indicates that PHCZ pollution is significantly influenced by industrial sources.

Comment (4):

Throughout the paper, the concepts of congener and homologue are messed up. A total of 11 congeners were measured in this work, and they are in three homologues – chlorinated, brominated, and chlorobromocarbazoles. The section starting in L162 has “congener” in the section title, but the content is mostly on homologues. In many places, the word “homologue” is used but individual congeners are focused.

Response:

Thank you for your comment. We have made corrections accordingly in the revised manuscript.

L159–160: Congener profiles and homologue patterns of PHCZs in matrices from industrial activities

L161–163: All target congeners were detected in i-PM material from COP, SAI, EAF, and MSWI (Supplementary Information 6), while specific congeners, especially chlorobromocarbazoles, could not be detected in most i-PM from other industries.

L190–194: Spearman analysis of the congener concentrations and environmental occurrences of PHCZs produced by the investigated industries (see Supplementary Information 6 and Supplementary Information 3) was conducted to understand the source and environmental characteristics of PHCZs (data processing in Supplementary Information 7).

L225–226: Comparison of congener and homologue patterns between i-PM and environmental samples

Comment (5):

The English is understandable, but lacks the concision and brevity. Some grammatical errors are present, especially towards the later section (e.g. Method). There are many redundant sentences and expressions in this paper. Significant efforts are needed to meet the criteria for technical writing, especially for publication in highly ranked journals.

Response:

Thank you for your comment. We have polished our manuscript using a professional editing company recommended by Springer Nature and ACS. Certificate of editing is displayed below.

Comment (6):

L43: References 2-5 appear to be randomly selected. Randomly picking of references should be avoided. Suggest removing Refs 1-5 or moving them to specific places in the paper where they fit more specifically.

Response:

Thank you for your comment. We have deleted the references accordingly.

Comment (7):

L40-53: Given the focus of this paper is on PHCZ sources, the entire first paragraph about toxicity can be deleted for brevity and to avoid misunderstanding on the focus of this paper.

Response:

Thank you for your comment. We have deleted the paragraph about toxicity.

L39: PHCZs, as emerging dioxin-like compounds (DLCs)³⁰

Comment (8):

L60: Two papers should be cited along with Ref 25: <https://dx.doi.org/10.1021/es503936u> and <https://doi.org/10.1021/acsestwater.2c00191>. Please also replace “concentrations of 38 ng/g led to” with “a median concentration of 23.7 ng/g in surface grabs, and”. The number 23.7 can be found in Li et al. 2022 Table S4. This number does not “lead” to the total load of 3354 tonnes in Table S6.

Response:

Thank you for your comment. We have added the references at L60 (L49 in revised manuscript) and changed ‘26 PHCZs with concentrations of 38 ng/g led to an estimated > 3000 tonnes in the sediments’ to ‘26 PHCZs with a median concentration of 23.7 ng/g in surface grabs, leading to an estimated >3000 tonnes in sediments’ at L59 (L47–48 in revised manuscript).

L48–51: An investigation of the pollution status of the Great Lakes revealed the existence of 26 PHCZs with a median concentration of 23.7 ng/g in surface grabs, leading to an estimated >3000 tonnes in sediment, which is orders of magnitude greater than those of PCBs and PBDEs^{29,31,32}.

Comment (9):

L60-63: Delete these two sentences. Although nothing seems wrong, ambiguous words such as “huge” lower the quality of writing.

Response:

Thank you for your comment. We have deleted the two sentences.

Comment (10):

L82: Move the comma to after “inadequate”.

Response:

Thank you for your comment. The comma in ‘current research on source identification is woefully inadequate, and knowledge to date’ has been removed following editing.

L69-70: Thus, current research on source identification is woefully inadequate.

Comment (11):

L86-93: Suggest deleting these redundant statements.

Response:

Thank you for your comment. We have deleted these statements.

Comment (12):

L95: “particulate matter”, not “matters” (no “s”). Please check throughout the paper.

Response:

Thank you for your comment. We have changed ‘particulate matters’ into ‘particulate matter’ throughout the paper.

Comment (13):

L96: Suggest ending the sentence after “liquids” and starting a new sentence “The results were examined for links and correlations with reported congener profiles in environmental samples, with the aim of identifying sources”. SI-2 and SI-4 should be mentioned in the Result section, not here.

Response:

Thank you for your comment. We have modified these sentences and marked them in the revised manuscript.

L80–84: Industrial fine particulate matter (i-PM) and chemical bottom liquids were analyzed to identify the 11 PHCZ congeners frequently detected in the environment (Supplementary Information 2). PHCZ congener profiles of industrial samples were compared with those of environmental samples (Supplementary Information 3).

L190–194: Spearman analysis of the congener concentrations and environmental occurrences of PHCZs produced by the investigated industries (see Supplementary Information 6 and Supplementary Information 3) was conducted to understand the source and environmental characteristics of PHCZs (data processing in Supplementary Information 7).

Comment (14):

L107-112: Suggest dividing this long sentence into two. One is “The industries included”. Then, “Data obtained are shown in Fig. 1a”.

Response:

Thank you for your comment. We have modified these sentences and marked them in the revised manuscript.

L90–92: Material samples potentially with PHCZs originated in the following industries: OC, CK, MSWI, HWI, COP, EAF, IOS, PCu, SAl, SCu and SZn. Data obtained are shown in Fig. 1.

Comment (15):

L163-176: Suggest deleting these text “Emissions So,” which sound redundant and help nothing.

Response:

Thank you for your comment. We have deleted these sentences.

Comment (16):

L171-172: Suggest changing “some of them contained a large amount of PHCZs” to “the PM from some of these industries had high concentrations of total PHCZs”.

Response:

Thank you for your comment. We have changed this expression.

Comment (17):

L195: The starting sentence is incorrect. This study is NOT the first discovering industrial sources of PHCZs. This paragraph is no use and should be deleted for brevity.

Response:

Thank you for your comment. We have deleted this paragraph.

Comment (18):

Fig 1: SA1 < COP in concentration, but SA1 > COP in TEQ. The reason might be the higher content in SA1 of 2367-CCZ and 27-CCZ which are more toxic. If true, a brief explanation should be added to main text L153-155.

Response:

Thank you for your comment. We agree with the reviewer's conclusion and have added a brief explanation at L150–152.

L150–152: Compared with i-PM from COP, i-PM produced from SCu-7 and SA1-1 have lower PHCZ concentrations but higher contents of more toxic congeners, 1,3,6,8-CCZ and 2,3,7,8-CCZ, which leads to a higher TEQc (Supplementary Information 6).

Comment (19):

Fig 2: Please deleting half of the data which repeat the other half. In the caption, delete words “homologue”. More importantly, are the original concentration data used for Spearman analyses in (b) from this study? Please specify the source of data in the caption. Also, the word “fly ash” is used here and in Supplementary Information-2. Do you consider i-PM and fly ash the same thing?

Response:

Thank you for your comment. We have modified the figure as suggested. The citations of environmental investigations have been added to the supplementary information 3. In our research, we obtained SEM images of our samples and the results indicated the particle diameters are less than 2.5 µm. Relevant results are provided in Supplementary Information 11.

Supplementary Information 3

Environmental occurrences of PHCZs cited in this research^{7,9–18}

Environmental occurrences and references	Location
Sedi-1 ⁹	Northern Germany
Sedi-2 ⁷	San Francisco Bay, USA
Sedi-3 ¹¹	Lake Tai, China
Sedi-4 ¹²	Zhou Shan, China
Sedi-5 ¹³	Qingdao, China
Soil-1 ¹²	Taizhou, China
Soil-2 ¹⁴	Zhoucheng, China
Soil-3 ¹⁵	Northeast Provinces, China
Soil-4 ¹⁵	Northeast Provinces, China
Soil-5 ¹⁵	Northeast Provinces, China
Soil-6 ¹⁵	Northeast Provinces, China
Soil-7 ¹⁵	Northeast Provinces, China
Soil-8 ¹⁶	Tibetan Plateau, China
Soil-9 ¹⁷	Hangzhou, China
Dust-1 ¹⁸	Derenda, Teltow, Germany
Dust-2 ¹⁰	Hangzhou, China

Supplementary Information 11

Scanning electron microscopy of industrial fine particulate matter samples

Comment (20):

Fig 5b: The unit is “g/t”, which is not easily understood. Suggest adding explanations in the caption, “grams of i-PM emitted per tonne of production capacity”, if correct.

Response:

Thank you for your comment. We have modified the caption accordingly and marked it in the **Figure File**.

Figure 5 Selected industries' (A) The production/incineration amount of several important industries⁶¹⁻⁶⁶, (B) i-PM emission factors^{40,67}, (C) The estimated annual global emission of PHCZs from the industries

Comment (21):

Fig 6: The unit for the numbers is “g” (gram), which is incorrect. Please clarify it is grams per year, per day, or per hour? If year, please add “annual” in the caption. Also, which year should be specified – 2010 or 2020 or another?

Response:

Thank you for your comment. We have modified the unit and added the year (2019) and marked it in the **Figure File**.

Figure 6 The preliminary estimation of annual global emission of PHCZs (grams/ milligrams per year) through COP, IOS, EAF and MSWI based on 2019 production data

Comment (22):

Table 1: All concentrations are reported with precision to 1 pg/g. Some larger numbers have six significant figures! Please re-consider the precision level of the data reported in this paper.

Response:

Thank you for your comment. We have reconsidered the significant figures, and we have placed Table 1 into the supplementary information.

References:

1. Ba, T. E. *et al.* Estimation and congener-specific characterization of polychlorinated naphthalene emissions from secondary nonferrous metallurgical facilities in China. *Environ. Sci. Technol.* **44**, 2441–2446 (2010).
2. Wu, X. *et al.* Thermochemical formation of polychlorinated dibenzo-p-dioxins and dibenzofurans on the fly ash matrix from metal smelting sources. *Chemosphere* **191**, 825–831 (2018).
3. Chen, Z., Lin, X., Lu, S., Li, X. & Yan, J. Suppressing formation pathway of PCDD/Fs by S-N-containing compound in full-scale municipal solid waste incinerators. *Chem. Eng. J.* **359**, 1391–1399 (2019).
4. Sun, H. Review on the extraction process of refined anthracene/carbazole in tar. *COAL Convers.* **21**, 29–32 (1998).
5. Riddell, N. *et al.* Characterization and Biological Potency of Mono- to Tetra-Halogenated Carbazoles. *Environ. Sci. Technol.* **49**, 10658–10666 (2015).
6. Ministry of Ecology and Environment of the People’s Republic of China. Handbook of Emission Sources Inventory Survey, Pollution Discharge Calculation Methods, and Coefficients. (2021).
7. Wu, Y., Tan, H., Sutton, R. & Chen, D. From Sediment to Top Predators: Broad Exposure of Polyhalogenated Carbazoles in San Francisco Bay (U.S.A.). *Environ. Sci. Technol.* **51**, 2038–2046 (2017).
8. Wang, G. *et al.* Occurrence and exposure risk evaluation of polyhalogenated carbazoles (PHCZs) in drinking water. *Sci. Total Environ.* **750**, 141615 (2021).
9. Chen, W. L. *et al.* Quantitative determination of ultra-trace carbazoles in sediments in the coastal environment. *Chemosphere* **150**, 586–595 (2016).
10. Zhou, H., Dong, X., Zhao, N., Zhao, M. & Jin, H. Polyhalogenated carbazoles in indoor dust from Hangzhou, China. *Sci. Total Environ.* **859**, 159971 (2023).
11. Wu, Y., Qiu, Y., Tan, H. & Chen, D. Polyhalogenated carbazoles in sediments from Lake Tai (China): Distribution, congener composition, and toxic equivalent evaluation. *Environ. Pollut.* **220**, 142–149 (2017).
12. Zhou, Y. *et al.* Method development for analyzing ultratrace polyhalogenated carbazoles in soil and sediment. *Ecotoxicol. Environ. Saf.* **182**, 109470 (2019).
13. Jia, Y. *et al.* Sediment-water distribution and potential sources of polyhalogenated carbazoles in a coastal river locating at a north metropolis, China. *Mar. Pollut. Bull.* **189**, 114790 (2023).
14. Tao, W. *et al.* Determination of polyhalogenated carbazoles in soil using gas chromatography-triple quadrupole tandem mass spectrometry. *Sci. Total Environ.* **710**, (2020).
15. Liu, M. *et al.* Occurrence and potential sources of polyhalogenated carbazoles in farmland soils from the Three Northeast Provinces, China. *Sci. Total Environ.* **799**, 149459 (2021).
16. Liu, M. *et al.* Occurrence and distribution of polyhalogenated carbazoles in eastern Tibetan Plateau soils along the slope of Mt. Qionglai. *Chemosphere* **298**, 134200 (2022).
17. Su, Q. *et al.* Soil to earthworm bioaccumulation of polyhalogenated carbazoles and related compounds: Lab and field tests. *Environ. Pollut.* **316**, (2023).
18. Fromme, H. *et al.* Occurrence of carbazoles in dust and air samples from different locations in Germany. *Sci. Total Environ.* **610–611**, 412–418 (2018).
19. Wu, Y. *et al.* Bioaccumulation and Spatiotemporal Trends of Polyhalogenated Carbazoles in Great Lakes Fish from 2004 to 2016. *Environ. Sci. Technol.* **52**, 4536–4545 (2018).
20. Parette, R. *et al.* Response to the comment on ‘Halogenated indigo dyes: A likely source of 1,3,6,8-tetrabromocarbazole and some other halogenated carbazoles in the environment’. *Chemosphere* **150**, 414–415 (2016).
21. International Council of Chemical Association. <https://icca-chem.org>.

22. Mumbo, J., Lenoir, D., Henkelmann, B. & Schramm, K. W. Enzymatic synthesis of bromo- and chlorocarbazoles and elucidation of their structures by molecular modeling. *Environ. Sci. Pollut. Res.* **20**, 8996–9005 (2013).
23. UNEP. Toolkit for identification and quantification of releases of dioxins, furans and other unintentional POPs under Article 5 of the Stockholm Convention on Persistent Organic Pollutants. 241–287 (2013).
24. Elmabruk, A. *et al.* Design, Synthesis, and Pharmacological Characterization of Carbazole Based Dopamine Agonists as Potential Symptomatic and Neuroprotective Therapeutic Agents for Parkinson's Disease. *ACS Chem. Neurosci.* **10**, 396–411 (2019).
25. Chau, N. Y., Ho, P. Y., Ho, C. L., Ma, D. & Wong, W. Y. Color-tunable thiazole-based iridium(III) complexes: Synthesis, characterization and their OLED applications. *J. Organomet. Chem.* **829**, 92–100 (2017).
26. Yuan, Z. W.; He, C.; Shi, Q.; Xu, C. M.; Li, Z. S.; Wang, C. Z.; Zhao, H. Z.; Ni, J. R. Molecular Insights into the Transformation of Dissolved Organic Matter in Landfill Leachate Concentrate during Biodegradation and Coagulation Processes Using ESI FT-ICR MS. *Environ. Sci. Technol.* **51**, 8110–8118 (2017).
27. Bianco, A.; Deguillaume, L.; Vaitilingom, M.; Nicol, E.; Baray, J. L.; Chaumerliac, N.; Bridoux, M., M. Characterization of Cloud Water Samples Collected at the Puy de Dome (France) by Fourier Transform Ion Cyclotron Resonance Mass Spectrometry. *Environ. Sci. Technol.* **52**, 10275–10285 (2018).
28. Sakai, S.-I., Hayakawa, K., Takatsuki, H. & Kawakami, I. Dioxin-like PCBs Released from Waste Incineration and Their Deposition Flux. *Environ. Sci. Technol.* **35**, 3601–3607 (2001).
29. Guo, J. *et al.* Spatial and Temporal Trends of Polyhalogenated Carbazoles in Sediments of Upper Great Lakes: Insights into Their Origin. *Environ. Sci. Technol.* **51**, 89–97 (2017).
30. Altarawneh, M. & Dlugogorski, B. Z. Formation and chlorination of carbazole, phenoxazine, and phenazine. *Environ. Sci. Technol.* **49**, 2215–2221 (2015).
31. Guo, J. *et al.* Polyhalogenated carbazoles in sediments of Lake Michigan: A new discovery. *Environ. Sci. Technol.* **48**, 12807–12815 (2014).
32. Li, A. *et al.* Polyhalogenated Carbazoles in Sediments of Lower Laurentian Great Lakes and Regional Perspectives. *ACS ES&T Water* **2**, 1544–1554 (2022).

Reviewer #1 (Remarks to the Author):

Generally, the authors have addressed all of my concerns with the original manuscript. Moreover, the number of investigated full-scale industrial plants has been greatly increased in the revised manuscript. The number of over 120 industrial plants is very large. The authors have explained the representative of the investigated industrial sources in the revised manuscript. The authors have also performed the comparison of emissions of PHCZs from different industrial sources and make reasonable explanations. In my opinion, the revised version is perfect and I recommend the revised manuscript to be accepted for publication.

Reviewer #3 (Remarks to the Author):

Compared with the original version, this revision is a significant improvement. The number of samples is increased from 26 to 122, with at least 6 samples for each of the 13 industries. The data analysis appears to be thorough, and the results are excellent presented in Figures and Tables. Overall, this work contributes significantly to the current environmental PHCZ research.

When citing references, it is important to avoid exaggerative generalization of specific findings. More cautions are needed in this paper, for example: (1) Ref 18 investigated two polymers with 2,7- and 3,6- linkages, thus their finding of 2,7- and 3,6- BCZs is limited to these two polymers. However, the sentence in L60-61 generalized the findings of ref 18, as if only 2,7- and 3,6- BCZs could be produced from the production of all photoelectric materials. (2) Similarly, does the statement in L64-65 apply to all historical synthetic dyes, or only the specific dyes investigated by refs 17 and 19? (3) It is also hard to accept the over-simplified statement in L216-219 about the origin of the three PHCZ congeners, because neither ref 14 nor ref 7 provided direct evidence of PHCZ formation in the implied sources. Over-simplifications of research findings may reduce the dimensionality and complexity of the reality, and could negatively affect data interpretation and future study designs.

Because of the (still) limited number of samples (< 6) in each industry category, I suggest reducing the discussion on Spearman correlation results (Fig 3) and greatly shortening L190-224 for brevity. The discussions around 3,6-CCZ (L236-251) also need additional caution. The mutual interferences of 3,6-CCZ and pesticide DDT in laboratory analysis using GC-MS and GC-MS/MS have been reported (Reischl et al. 2005, Chlorcarbazole in böden. Umweltwissenschaften und Schadstoff-Forschung, 17, 197; Zhou et al. 2019 Ecotoxicology and Environmental Safety, 182, 109470). Given the relatively high DDT concentrations in soils and sediments of China, some published data on 3,6-CCZ may need to be re-evaluated.

Minor comments and suggestions

Title: Suggest changing "new" to "industrial". It might also be better to replace "estimation" with "preliminary assessment".

L42: Suggest changing "equal" to "comparable".

L53: "Cl-/Br-" is confusing. Do you mean chloride and bromide ions, or their ratio, or something else?

L59: Refs 17 and 18 are not considered "recently". Suggest deleting "as well as materials", and moving statements about indigo dye and photoelectric materials to later sentences.

L60: The word "However" is not needed and confusing. Also, based on how many and which references, 3,6-BCZ is considered low in environmental media?

L86: Move "for the first time" to immediately after "mapped" and before the comma. This study is certainly not the first one using emission factor method.

L92: Suggest adding "11" as a subscript for the sigma in all "sigmaPHCZs" in this paper, to

indicate clearly how many congeners are included. This term should be used whenever appropriate; for example, replace "total PHCZs" (L233) and "11 PHCZs" (L240) with "sigma11PHCZs".

L100: Delete "but unrecognized".

L101: Change "can be obtained from i-PM" to "were found in i-PM samples".

L152: "2,3,7,8-" is likely to be a typo. This congener is not included in this study.

L154-158: Strongly suggest deleting this sentence, which is irrelevant to this study.

L164: Change "could not be" to "were not".

L167: Suggest changing "occupy a larger area with proportions more than 70% in" to "count for more than 70% in samples from".

L174-174: In the SI version I downloaded, 2,3,6,7-CCZ is missing from SI-6.

L275: Switch "investigated" and "13".

SI-5: "Match sore" is probably "Match score".

Response to reviewers' comments

Reviewer #1:

General comments:

Generally, the authors have addressed all of my concerns with the original manuscript. Moreover, the number of investigated full-scale industrial plants has been greatly increased in the revised manuscript. The number of over 120 industrial plants is very large. The authors have explained the representative of the investigated industrial sources in the revised manuscript. The authors have also performed the comparison of emissions of PHCZs from different industrial sources and make reasonable explanations. In my opinion, the revised version is perfect and I recommend the revised manuscript to be accepted for publication.

Response:

We greatly appreciate your recognition on the representative of the investigated industrial sources and the discussions in our revised manuscript. Your constructive and insightful suggestions and comments play a strong role in the improvement in our revised manuscript. Thanks again!

Reviewer #3:

General comments:

Compared with the original version, this revision is a significant improvement. The number of samples is increased from 26 to 122, with at least 6 samples for each of the 13 industries. The data analysis appears to be thorough, and the results are excellent presented in Figures and Tables. Overall, this work contributes significantly to the current environmental PHCZ research.

Response:

We sincerely appreciate your recognition on the improvement in our revised manuscript. Thanks for the constructive comments. We have carefully revised our manuscript based on your comments.

Specific comments:

Comment (1):

When citing references, it is important to avoid exaggerative generalization of specific findings. More cautions are needed in this paper, for example: (1) Ref 18 investigated two polymers with 2,7- and 3,6- linkages, thus their finding of 2,7- and 3,6- BCZs is limited to these two polymers. However, the sentence in L60-61 generalized the findings of ref 18, as if only 2,7- and 3,6- BCZs could be produced from the production of all photoelectric materials. (2) Similarly, does the statement in L64-65 apply to all historical synthetic dyes, or only the specific dyes investigated by refs 17 and 19? (3) It is also hard to accept the over-simplified statement in L216-219 about the origin of the three PHCZ congeners, because neither ref 14 nor ref 7 provided direct evidence of PHCZ formation in the implied sources. Over-simplifications of research findings may reduce the dimensionality and complexity of the reality, and could negatively affect data interpretation and future study designs.

Response:

Thanks for your reviews and insightful comments. We have reviewed all the findings in the references in this manuscript to avoid exaggerative generalization. Detailed revisions are highlighted in revised manuscript and shown below.

L55-59: Recently, PHCZs have been found in disinfection of drinking water¹⁰. The production of halogenated indigo dyes¹⁷ and specific photoelectric materials¹⁸ are also considered as potential sources of PHCZs. 2,7- and 3,6- halogenated carbazoles are possible intermediates of two polymers present in electronic devices, which have not been detected directly in these industrial samples.

L60-62: The synthesis of halogenated indigo dyes is also considered a source of PHCZ emission^{19,20}, although one modern indigo dye, 5,5',7,7'-tetrabromoindigo was proved not to contain PHCZs¹⁹.

L196-199: The result is consistent with findings reported by Guo et al¹⁴, but more field studies are needed to further confirm pollution contribution of these

investigated sources as well as natural sources and other unexplored artificial sources in specific areas.

Comment (2):

Because of the (still) limited number of samples (< 6) in each industry category, I suggest reducing the discussion on Spearman correlation results (Fig 3) and greatly shortening L190-224 for brevity. The discussions around 3,6-CCZ (L236-251) also need additional caution. The mutual interferences of 3,6-CCZ and pesticide DDT in laboratory analysis using GC-MS and GC-MS/MS have been reported (Reischl et al. 2005, Chlorcarbazole in böden. Umweltwissenschaften und Schadstoff-Forschung, 17, 197; Zhou et al. 2019 Ecotoxicology and Environmental Safety, 182, 109470). Given the relatively high DDT concentrations in soils and sediments of China, some published data on 3,6-CCZ may need to be re-evaluated.

Response:

Thanks for your comments. We totally agree with the Reviewer's suggestions on the discussion on Spearman correlation results as well as origin of 3,6-CCZ. We shorten the part of Spearman correlations for brevity. Considering the separation of 3,6-CCZ and DDT in GC system as shown in figure below from Zhou's report (Zhou et al. 2019 Ecotoxicology and Environmental Safety, 182, 109470), as well as clean-up processes in laboratory analysis, the published data about 3,6-CCZ in soil and sediment is accepted in this preliminary comparison of characteristics of PHCZs from industrial sources and these in the environment. But we also point out the possible influence generated by impurities in laboratory analysis. Detailed revisions are highlighted in revised manuscript and shown below.

Redacted

(Zhou et al. 2019 Ecotoxicology and Environmental Safety, 182, 109470)

L185-189: As shown in Fig. 3 (red and blue indicate positive and negative correlations, respectively), certain congeners display a strong positive correlation with other congeners in SAI, MSWI, and CK. In the other industries, such as COP, SZn, and IOS, correlations exist between minority of congeners including 3-CCZ, 3-BCZ, 3,6-CCZ, and 1,3,6,8-CCZ.

L193-201: The proportion of 3,6-CCZ in i-PM samples is high but the correlations between 3,6-CCZ and other congeners in the environment and industries differ, suggesting that anthropogenic and natural sources might jointly contribute to 3,6-CCZ pollution in sediment. The result is consistent with findings reported by Guo et al¹⁴, but more field studies are needed to further confirm pollution contribution from these investigated sources as well as natural sources and other unexplored artificial sources in specific areas. The mutual interferences of 3,6-CCZ and pesticide DDT^{39,40} in laboratory analysis should also be taken into consideration when the discrepancy appeared.

Comment (3):

Title: Suggest changing “new” to “industrial”. It might also be better to replace “estimation” with “preliminary assessment”.

Response:

Thanks for your comments. We have changed the relevant part in revised manuscript. Detailed revisions are highlighted in revised manuscript and shown below.

Title: Polyhalogenated carbazoles: Identification of industrial sources and preliminary assessment of global emissions

Comment (4):

L42: Suggest changing “equal” to “comparable”.

Response:

Thanks for your comments. We have changed the relevant part in revised manuscript. Detailed revisions are highlighted in revised manuscript and shown below.

L39: Their concentrations are comparable to or even exceed those of some traditional DLCs.

Comment (5):

L53: “Cl-/Br-“ is confusing. Do you mean chloride and bromide ions, or their ratio, or something else?

Response:

Thanks for your comments. ‘Cl-/Br-’ is here to represent chloride and bromide ions. We have changed the relevant part in revised manuscript. Detailed revisions are highlighted in revised manuscript and shown below.

L50: Mumbo et al.¹⁶ confirmed that in the presence of H₂O₂ and chloride and bromide ions, fungi can bio-transform carbazole (CZ) to PHCZs.

Comment (6):

L59: Refs 17 and 18 are not considered “recently”. Suggest deleting “as well as materials”, and moving statements about indigo dye and photoelectric materials to later sentences.

Response:

Thanks for your comments. We have changed the relevant part in revised manuscript. Detailed revisions are highlighted in revised manuscript and shown below.

L55-57: Recently, PHCZs have been found in disinfection of drinking water¹⁰. The production of halogenated indigo dyes¹⁷ and specific photoelectric materials¹⁸ are also considered as potential sources of PHCZs.

Comment (7):

L60: The word “However” is not needed and confusing. Also, based on how many and which references, 3,6-BCZ is considered low in environmental media?

Response:

Thanks for your comments. The 3,6-BCZ accounts for a relatively high proportion of 11 PHCZs in some research, such as 3,6-BCZ in sediments from Lake Tai (Wu et al. Environmental Pollution 220 (2017) 142-149). We have reviewed relevant articles and changed the relevant part in revised manuscript. Detailed revisions are highlighted in revised manuscript and shown below.

L58-59: 2,7- and 3,6- halogenated carbazoles are possible intermediates of two polymers present in electronic devices, which have not been detected directly in these industrial samples.

Comment (6):

L86: Move “for the first time” to immediately after “mapped” and before the comma. This study is certainly not the first one using emission factor method.

Response:

Thanks for your comments. We have changed the relevant part in revised manuscript. Detailed revisions are highlighted in revised manuscript and shown below.

L82-84: Annual global PHCZ emissions from studied industrial activities producing PHCZs at relatively high concentrations were also estimated and mapped using the emission factor method.

Comment (7):

L92: Suggest adding “11” as a subscript for the sigma in all “sigmaPHCZs” in this paper, to indicate clearly how many congeners are included. This term should be used whenever appropriate; for example, replace “total PHCZs” (L233) and “11 PHCZs” (L240) with “sigma₁₁PHCZs”.

Response:

Thanks for your comments. We have changed all the relevant part in revised manuscript.

Comment (8):

L100: Delete “but unrecognized”.

Response:

Thanks for your comments. We have changed the relevant part in revised manuscript. Detailed revisions are highlighted in revised manuscript and shown below.

L98: making COP a significant industrial source of PHCZs.

Comment (9):

L101: Change “can be obtained from i-PM” to “were found in i-PM samples”.

Response:

Thanks for your comments. We have changed the relevant part in revised manuscript. Detailed revisions are highlighted in revised manuscript and shown below.

L99: Relatively high Σ 11PHCZs were found in i-PM samples from IOS, SAI, SCu, SZn, MSWI, and HWI, ranging from 8 ng/g to 60 ng/g.

Comment (10):

L152: “2,3,7,8-“ is likely to be a typo. This congener is not included in this study.

Response:

Thanks for your comments. We have changed the relevant part in revised manuscript. Detailed revisions are highlighted in revised manuscript and shown below.

L149: 1,3,6,8-CCZ and 2,3,6,7-CCZ, which leads to a higher TEQ_c

Comment (11):

L154-158: Strongly suggest deleting this sentence, which is irrelevant to this study.

Response:

Thanks for your comments. We have deleted the sentence.

Comment (12):

L164: Change “could not be” to “were not”.

Response:

Thanks for your comments. We have changed the relevant part in revised manuscript. Detailed revisions are highlighted in revised manuscript and shown below.

L157: especially chlorobromocarbazoles, were not detected in most i-PM from other industries.

Comment (13):

L167: Suggest changing “occupy a larger area with proportions more than 70% in” to “count for more than 70% in samples from”.

Response:

Thanks for your comments. We have changed the relevant part in revised manuscript. Detailed revisions are highlighted in revised manuscript and shown below.

L160: As shown in Fig. 2, compared with other halogen-substituted carbazoles, chlorinated carbazoles count for more than 70% in samples from EAF, HWI, PCu, SCu, and SZn.

Comment (14):

L174-174: In the SI version I downloaded, 2,3,6,7-CCZ is missing from SI-6.

Response:

Thanks for your comments. We have changed the relevant part in revised supplementary information.

Comment (15):

L275: Switch “investigated” and “13”.

Response:

Thanks for your comments. We have changed the relevant part in revised manuscript. Detailed revisions are highlighted in revised manuscript and shown below.

L261: Although PHCZs are generally released from the 13 investigated industries,

Comment (16):

SI-5: “Match sore” is probably “Match score”.

Response:

Thanks for your comments. We have changed the relevant part in revised supplementary information.